# InstructSAM: A Training-Free Framework for Instruction-Oriented Remote Sensing Object Recognition

**Yijie Zheng, Weijie Wu**
Aerospace Information Research Institute,
Chinese Academy of Sciences
University of Chinese Academy of Sciences
`{zhengyijie23,wuweijie25}@mails.ucas.ac.cn`

**Qingyun Li**
Harbin Institute of Technology
`21b905003@stu.hit.edu.cn`

**Xuehui Wang**
Shanghai Jiao Tong University
`wangxuehui@sjtu.edu.cn`

**Xu Zhou, Aiai Ren, Jun Shen**
University of Wollongong
`{xz572,ar243,jshen}@uowmail.edu.au`

**Long Zhao, Guoqing Li**✉
Aerospace Information Research Institute,
Chinese Academy of Sciences
`{zhaolong,ligq}@aircas.ac.cn`

**Xue Yang**
SAIS, Shanghai Jiao Tong University
`yangxue-2019-sjtu@sjtu.edu.cn`

## Abstract

Language-guided object recognition in remote sensing imagery is crucial for large-scale mapping and automated data annotation. However, existing open-vocabulary and visual grounding methods rely on explicit category cues, limiting their ability to handle complex or implicit queries that require advanced reasoning. To address this issue, we introduce a new suite of tasks, including Instruction-Oriented Object Counting, Detection, and Segmentation (InstructCDS), covering open-vocabulary, open-ended, and open-subclass scenarios. We further present EarthInstruct, the first InstructCDS benchmark for earth observation. It is constructed from two diverse remote sensing datasets with varying spatial resolutions and annotation rules across 20 categories, necessitating models to interpret dataset-specific instructions. Given the scarcity of semantically rich labeled data in remote sensing, we propose InstructSAM, a training-free framework for instruction-driven object recognition. InstructSAM leverages large vision-language models to interpret user instructions and estimate object counts, employs SAM2 for mask proposal, and formulates mask-label assignment as a binary integer programming problem. By integrating semantic similarity with counting constraints, InstructSAM efficiently assigns categories to predicted masks without relying on confidence thresholds. Experiments demonstrate that InstructSAM matches or surpasses specialized baselines across multiple tasks while maintaining near-constant inference time regardless of object count, reducing output tokens by 89% and overall runtime by over 32% compared to direct generation approaches. We believe the contributions of the proposed tasks, benchmark, and effective approach will advance future research in developing versatile object recognition systems. The code is available at `https://VoyagerXvoyagerx.github.io/InstructSAM/`.

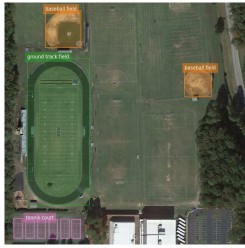 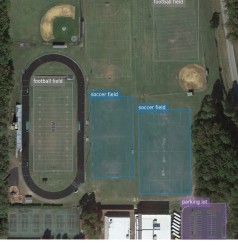 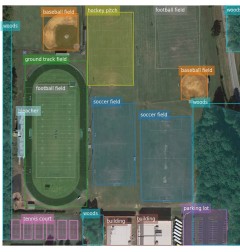 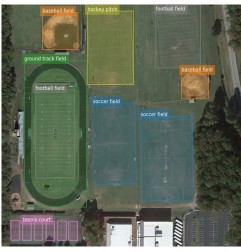

| (a) Close-set | (b) Open-vocabulary | (c) Open-ended | (d) Open-subclass |

Figure 1: Comparison of four task settings in object detection and segmentation. The sample is from DIOR Dataset [34] (a) Close-set: Standard annotation with categories defined in the DIOR dataset. (b) Open-vocabulary: Instruction specifies which categories to detect and segment (e.g., "soccer field", "football field", "parking lot"). (c) Open-ended: Instruction requires detection and segmentation of all visible objects without category specification. (d) Open-subclass: Instruction targets all objects within a super-category (e.g., "sports fields").

# 1  Introduction

Object recognition in remote sensing imagery captures a vast array of objects and phenomena across diverse environments, providing rich information for supporting achieving the Sustainable Development Goals issued by the United Nations [59, 3], such as wildlife monitoring [82, 96], poverty estimation [1, 50], and disaster response [18]. The recent advent of powerful vision-language models (VLMs), such as CLIP [61], has ushered in a new era of remote sensing oriented open-vocabulary object recognition algorithms (e.g. detection [38, 56] and segmentation [21]). However, existing open-vocabulary approaches predominantly rely on explicit category cues, which restrict their capability to handle complex or implicit queries that demand advanced reasoning and contextual understanding. In other words, the rich diversity of visible objects in remote sensing images, due to the bird's-eye view, predicates that any predefined, fixed list of categories is inevitably incomplete, limiting its practicality for open-ended analysis in the real world.

To address this issue, we expand the instruction-oriented object detection task [60] and introduce a novel suite of tasks—Instruction-Oriented Object Counting, Detection, and Segmentation (**InstructCDS**) that encompasses open-vocabulary, open-ended, and open-subclass settings, as illustrated in Figure 1. The InstructCDS task entails a more flexible and scalable interpretation beyond fixed category sets and comprehends the complex users' task requirements. We further present **Earth-Instruct**, the first benchmark for InstructCDS in earth observation. The benchmark is constructed from two generic remote sensing object datasets covering 20 categories with different annotation rules and spatial resolution. EarthInstruct guides models to comprehend complex user instructions beyond the predefined three settings.

Recent advancements in VLMs have demonstrated impressive performance in object detection [45, 40, 60, 57], semantic segmentation [89, 11, 94], visual grounding [2, 7] and reasoning-based segmentation [66, 29] within the natural image domain. However, transferring these methods to remote sensing imagery presents several challenges. **First**, direct inference leads to significant accuracy degradation due to the domain gap between natural and aerial images [14]. **Second**, most existing remote sensing open-vocabulary detection [38, 39, 90, 31] and segmentation [91, 88, 21] methods are trained on datasets with only a limited number of categories, restricting their generalization to diverse unseen categories. **Third**, conventional detectors [35, 10, 51] rely on a threshold to filter predicted bounding boxes, which is not obtainable in zero-shot scenarios.

To tackle these challenges, we decompose the instruction-oriented object detection and segmentation tasks into several tractable steps and propose a framework without task-specific training, **Instruct-SAM**. First, a large vision-language model (LVLM) is employed to interpret user instructions and predict **object categories** and **counts**, with prompts systematically designed to maximize model's capability. In parallel, SAM2 [63] is utilized to automatically generate **mask proposals**. Next, a CLIP model pre-trained on remote sensing images computes the **semantic similarity** between the predicted objects categories and mask proposals. We then formulate the object detection and segmentation as a **mask-label matching** problem, assigning predicted categories to mask proposals

by integrating semantic similarity and global counting constraints. By inherently integrating three powerful foundation models, InstructSAM achieves superior performance across multiple tasks compared to both generic and remote sensing-specific VLMs trained on large-scale object recognition data. Notably, the inference time of InstructSAM remains nearly constant with respect to the number of predicted objects, reducing output tokens by 89% and overall runtime by 32% compared to directly generating bounding boxes using Qwen2.5-VL [2] in the open-ended setting. Our work paves the way for scalable, instruction-driven remote sensing object detection and segmentation, eliminating the need for costly pre-training or manual threshold tuning. Furthermore, the training-free paradigm allows InstructSAM to recognize objects in natural images when equipped with generic CLIP models.

In summary, our contributions are as follows:

- We introduce the InstructCDS task, which challenges models to interpret user-provided natural language instructions and infer the number and locations of relevant objects.
- We construct EarthInstruct to benchmark InstructCDS in earth observation, covering open-vocabulary, open-ended, and open-subclass settings and counting, detection, and segmentation tasks.
- We develop InstructSAM, a training-free and confidence-free framework achieving near-constant inference time for the InstructCDS task.
- Experiments on the benchmarks demonstrate that InstructSAM matches close-set models in object counting and surpasses generic and remote sensing-specific models in open-vocabulary and open-ended object recognition.

## 2 Related work

### 2.1 Instruction-oriented object detection and segmentation

**Instruction-oriented methods**   Instruction-oriented object detection (IOD), first introduced in [60], includes four instruction settings: category-specified (open-vocabulary [80]) detection, detection of all objects (open-ended [40]), detection within a super-category (we name it open-subclass), and detection to achieve certain goals. Ins-DetCLIP [60] trains a detector to identify foreground objects and passes their features to a large language model (LLM) to generate categories based on user instructions. In addition to models specifically designed for the IOD task, Qwen2.5-VL [2], trained on multi-task instruction data, also showcases the ability of dense object detection. However, both approaches require extensive task-specific training data, and their inference times increase substantially with the number of objects.

**Open-vocabulary methods**   Large-vocabulary object detection and segmentation datasets [70, 17, 76] and visual grounding datasets [27, 23] enable various open-vocabulary learning approaches, including knowledge distillation [16] and region-text pre-training [26, 52, 45, 35]. Self-training with confidence threshold-filtered pseudo labels from image-text pairs (e.g., CC3M [5], WebLI2B [6]) further boosts the performance [51, 45, 93, 87]. However, the quality of pseudo labels is highly sensitive to the chosen threshold [51], and these methods require predefined object categories, limiting their flexibility in diverse scenarios.

**Open-ended methods**   GenerateU [40] first formulates the open-ended object detection (OED) problem. Concurrent works such as DetCLIPv3 [87], Florence-2 [83], and DINO-X [65] introduce generative frameworks that jointly predict object categories and bounding boxes using language models. However, constructing large-scale datasets with bounding box and caption pairs is resource-intensive. VL-SAM [41] proposes a training-free approach via attention as prompts, but its iterative mask refinement and multi-prompt ensemble strategies are computationally expensive.

### 2.2 Instruction-oriented remote sensing object detection and segmentation

Recent advances in VLMs [61, 44] have also enabled open-vocabulary learning in the remote sensing domain. Diverse semantic tags from OpenStreetMap [19] and those generated by LVLMs drive the development of contrastive language-image pre-training in remote sensing images [95, 77]. Following the generic open-vocabulary learning frameworks, methods for remote sensing open-vocabulary

detection [38, 79, 56, 22] and segmentation [91, 88, 21, 32] methods emerge. However, their human-annotated training data, being limited to a few dozen categories [34, 78, 73, 47], hinder generalization to out-of-distribution or zero-shot scenarios. Although some remote sensing LVLMs could support object recognition tasks like single-class object detection [28, 49, 25], visual grounding [54, 58], referring expression segmentation [97, 55], grounded conversation generation [68], and scene graph generation [49], they failed to follow complex reasoning instructions, such as open-vocabulary and open-subclass object detection. To annotate vast-vocabulary training data for remote sensing object detection, LAE-Label [56] employs a generic LVLM [8] to predict categories for cropped mask proposals. However, this approach loses global context for accurate category classification.

In contrast, our InstructSAM adopts a confidence-free paradigm, requires no task-specific pre-training or finetuning, and maintains near-constant inference time regardless of object counts.

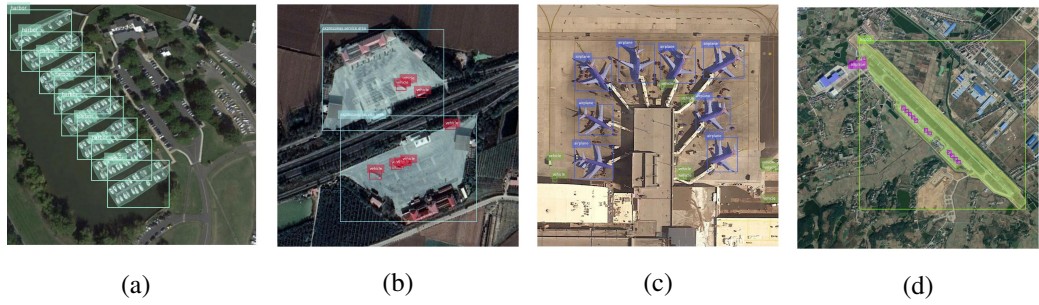

|        |        |        |        |
| :----: | :----: | :----: | :----: |
| (a)    | (b)    | (c)    | (d)    |

Figure 2: Annotation samples from NWPU-VHR-10 (a) and DIOR (b-d) illustrating rules that might differ from common sense. (a) Ships and vehicles are unannotated in low-resolution images (NWPU-VHR-10). (b) Expressway service areas separated by a road are treated as distinct instances (DIOR). (c, d) Airports are annotated only when fully visible (DIOR).

## 3 EarthInstruct, a benchmark for InstructCDS in remote sensing

### 3.1 Instruction setup

To enable practical applications in remote sensing such as large-scale mapping [1] and image annotation, we define three distinct settings for InstructCDS:

**1) Open-vocabulary**: Counting, detection, and segmentation with user-specified categories.

**2) Open-ended**: Counting, detection, and segmentation of all objects without specifying categories.

**3) Open-subclass**: Counting, detection, and segmentation of objects within a super-category.

We construct EarthInstruct using NWPU-VHR-10 [9] and DIOR [34] datasets, selected for their widespread use and diverse sensors, resolutions, and annotation rules. Critically, these dataset-specific annotation rules might deviate from common sense (e.g., excluding low-resolution vehicles) or exhibit semantic ambiguities (e.g., "bridge" vs. "overpass"), reflecting the original annotators' specific goals (Figure 2). Consequently, simple instructions like "count vehicles" would fail to capture the nuances required by dataset conventions or user intent. EarthInstruct therefore necessitates models to interpret detailed instructions that clarify target definitions and handle dataset-specific rules (e.g., "do not count vehicles in images with a spatial resolution lower than 1m."). To ensure fair evaluation aligned with dataset conventions and user requirements, prompts are designed accordingly, but image-specific prompts are prohibited to maintain scalability for large-area applications where prior content knowledge for each image is unavailable.

### 3.2 Evaluation metrics

**Multi-class object counting**  Standard counting metrics, such as Mean Absolute Error (MAE) and Root Mean Squared Error (RMSE), used in benchmarks like FSC-147 [62] and RSOC [15], inadequately capture nuanced multi-class evaluation. They cannot distinguish between over-counting and under-counting errors. Additionally, being unnormalized, they allow categories with larger counts to disproportionately skew the overall score when being averaged across classes.

To address these issues, we adopt precision, recall, and $F_1$-score, whereby offering normalized, per-class insights. We define per-image, per-class counting components as follows: let $C_{gt}$ denote the ground truth count and $C_{pred}$ the predicted count for a class in an image. Then, True Positives (TP) = $\min(C_{gt}, C_{pred})$, False Positives (FP) = $\max(0, C_{pred} - C_{gt})$ for over-counting, and False Negatives (FN) = $\max(0, C_{gt} - C_{pred})$ for under-counting. These definitions enable standard calculation of precision, recall, and $F_1$-score, aggregated across images per class and then averaged for a final score.

**Rethinking metrics for confidence-free detectors** Evaluating generative models like Florence-2 [83] or Qwen2.5-VL [2], which output detections without confidence scores, poses a challenge for standard metrics. Average Precision (AP) [12] relies on confidence scores to rank predictions and generate precision-recall curves. Without such ranking, standard AP is ill-defined. Furthermore, practical applications often involve filtering predictions with a fixed threshold, treating all remaining detections equally [1, 51].

To resolve these issues and ensure fair comparison, we adopt confidence-free metrics: mean $F_1$-score ($mF_1$) and mean Average Precision with no confidence ($mAP_{nc}$) [36]. $mF_1$ measures performance at a single operating point, suitable for fixed-threshold deployment. $mAP_{nc}$ adapts AP by assigning maximum confidence to all predictions. For confidence-free models, these metrics are computed directly (Results in Table 10). For conventional detectors (e.g., [64, 4]) that provide scores, when the confidence threshold is swept from 0 to 1 (step 0.02), the threshold maximizing $mF_1$ (using an IoU threshold of 0.5) across categories is selected, and the corresponding cusp score is reported.

**Evaluating open-ended and open-subclass settings** In open-ended and open-subclass settings, LVLMs may generate category names (e.g., "car") that differ textually from ground truth labels (e.g., "vehicle"). To handle this synonymy during evaluation, we adopt semantic similarity matching by following established protocols [40, 60]. Specifically, we encode generated categories and ground truth categories using the GeoRSCLIP [95] text encoder with the template ``a satellite image of a {category}''. A generated category name is deemed equivalent to a ground truth category if their embedding cosine similarity exceeds 0.95. This allows predicted objects associated with the generated name to be accurately evaluated against the matched ground truth category.

# 4   InstructSAM

Addressing the challenges of instruction-following, domain gaps, and threshold sensitivity in remote sensing object recognition, we propose a training-free framework named InstructSAM. It decomposes InstructCDS into three synergistic steps: instruction-based object counting using an LVLM, class-agnostic mask proposing via SAM2, and a novel counting-constrained mask-label matching procedure. This approach avoids the costly model training and threshold tuning, offering efficient and robust performance.

## 4.1   Instruction-oriented object counting with LVLM

As illustrated in Section 3, accurately interpreting user intent in remote sensing requires handling dataset-specific rules and semantic ambiguities, which the simple category prompts could fail to capture. We leverage state-of-the-art LVLMs (e.g., GPT-4o [24], Qwen2.5-VL [2]) for this task. Inspired by [20], we utilize structured prompts in JSON format, which allows easy integration of dataset-specific `Instructions` alongside the core `Task` (detailed in Appendix C). Given an image $I$ and a detailed prompt $P$, the LVLM acts as a counter, outputting the target categories $\{cat_j\}$ and their corresponding counts $\{num_j\}$ present in the image: $\{cat_j, num_j\}_{j=1}^{M} = \texttt{LVLM-Counter}(I, P)$.

## 4.2   Class-agnostic mask proposing

Concurrent with counting, SAM2 [63] is employed to generate high-quality, class-agnostic object masks for its strong generalization to remote sensing imagery [75, 81]. Using its automated mask generation mode prompted by a regular point grid, we obtain a dense set of masks proposals $\{mask_i\}_{i=1}^{N}$. To enhance recall for small objects, mask generation is also applied to image crops (detailed in Appendix C.4).

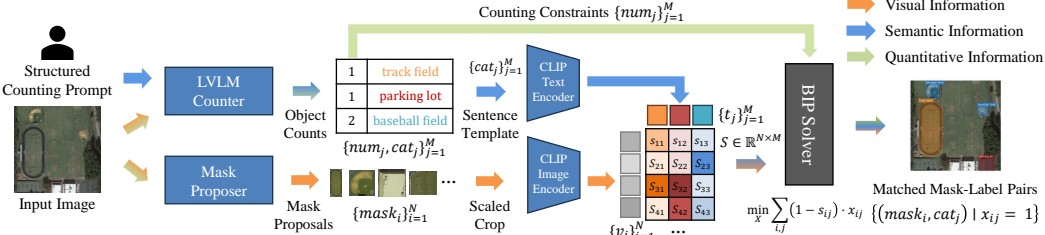

Figure 3: The InstructSAM framework. Given an input image and a structured counting prompt, the LVLM Counter extracts target categories $\{cat_j\}$ (semantic info) and their counts $\{num_j\}$ (quantitative info). Concurrently, the Mask Proposer generates mask proposals $\{mask_i\}$ (visual info). A CLIP model computes the similarity matrix $S$ between mask embeddings (from scaled crops) $\{v_i\}$ and category embeddings $\{t_j\}$. Finally, the Binary Integer Programming (BIP) Solver optimally assigns categories to masks by maximizing summed similarity, subject to the counting constraints, yielding the final recognition results.

## 4.3 Mask-label matching with counting constraints

A key innovation of InstructSAM is reframing object detection and segmentation as a constrained mask-label matching problem, by integrating the outputs from previous steps. Rather than using fragile confidence thresholds [10, 51], we utilize global counts $\{num_j\}$ derived from the LVLM to constrain the assignment of semantic labels $\{cat_j\}$ to visual mask proposals $\{mask_i\}$.

Given $N$ mask proposals and $M$ target categories with counts, we compute a semantic similarity matrix $S \in \mathbb{R}^{N \times M}$, where $s_{ij}$ represents the cosine similarity between the CLIP [95] image embedding of a patch cropped around $mask_i$ (scaled by 1.2 for context) and the text embedding of $cat_j$ (using the template ``a satellite image of a {category}''). We then seek a binary assignment matrix $X \in \{0, 1\}^{N \times M}$, where $x_{ij} = 1$ assigns $mask_i$ to $cat_j$, by solving the Binary Integer Programming (BIP) problem:

$$\min_{\mathbf{X}} \quad \sum_{i=1}^{N} \sum_{j=1}^{M} (1 - s_{ij}) \cdot x_{ij} \tag{1}$$

$$\text{s.t.} \quad \sum_{j=1}^{M} x_{ij} \leq 1, \qquad \forall i \in \{1, \ldots, N\} \tag{2}$$

$$\sum_{i=1}^{N} x_{ij} = num_j, \qquad \forall j \in \{1, \ldots, M\}, \quad \text{if } N \geq \sum_{j=1}^{M} num_j \tag{3}$$

$$\sum_{i=1}^{N} \sum_{j=1}^{M} x_{ij} = N, \qquad \text{if } N < \sum_{j=1}^{M} num_j \tag{4}$$

where constraint (2) ensures each mask is assigned to at most one category. Constraint (3) enforces that the number of assigned masks for each category matches the counts provided by the LVLM. Constraint (4) handles cases where the number of proposals is less than the total target count, ensuring all proposals are assigned.

As depicted in Figure 3, this BIP formulation elegantly fuses visual information, semantic information, and quantitative information. The visual information comes from the CLIP embeddings of mask proposals $\{v_i\}$, which contribute to $s_{ij}$. Semantic information is derived from the categories' sentence embeddings $\{t_j\}$, also contributing to $s_{ij}$. Quantitative information from object counts $\{num_j\}$ serves as constraints in (3). The problem is efficiently solvable using open-source BIP solvers like PuLP [53]. The resulting non-zero entries in $X$ define the final set of recognized objects $\{(mask_i, cat_j) | x_{ij} = 1\}$.

Table 1: Zero-shot performance comparison on EarthInstruct under open-vocabulary setting.

| Method | NWPU-VHR-10 | | | | DIOR | | | |
|---|---|---|---|---|---|---|---|---|
| | Cnt $F_1$ | Box $F_1$ | Mask $F_1$ | IoU | Cnt $F_1$ | Box $F_1$ | Mask $F_1$ | IoU |
| Grounding DINO [45] | 14.9 | 14.0 | - | - | 10.7 | 6.0 | - | - |
| OWLv2 [51] | 39.4 | 27.2 | - | - | 23.4 | 14.3 | - | - |
| Qwen2.5-VL [2] | 68.0 | 36.4 | - | - | 52.0 | 27.8 | - | - |
| GSNet [88] | - | - | 1.3 | 6.4 | - | - | 0.0 | 0.3 |
| SegEarth-OV [32] | - | - | 3.5 | 12.1 | - | - | 1.2 | 6.7 |
| InstructSAM-Qwen | 73.2 | 38.9 | 23.7 | 12.1 | 59.3 | 24.7 | 24.0 | 18.5 |
| InstructSAM-GPT4o | **83.0** | **41.8** | **26.1** | **14.8** | **79.9** | **29.1** | **28.1** | **20.2** |

# 5 Experiments

## 5.1 Implementation

We implement InstructSAM using GPT-4o-2024-11-20 [24] (short for InstructSAM-GPT4o) or Qwen2.5-VL-7B [2] (short for InstructSAM-Qwen) as the LVLM counter, SAM2-hiera-large [63] for mask proposal, and GeoRSCLIP-ViT-L [95] for similarity computation. For the open-vocabulary setting, we follow previous works [79, 21, 90] to split base and novel classes, and report $mF_1$, mean Insert-over-Union (mIoU), or $mAP_{nc}$. For open-subclass setting, we set two parent class "means of transport" and "sports fields". We compare InstructSAM with a wide range of models, whose training data and abilities are listed in Table 6.

## 5.2 Results on EarthInstruct

**Open-vocabulary setting** We report mean metrics across all classes for generic methods [45, 51, 2] and remote sensing open-vocabulary segmentation models [88, 32] trained on broader vocabularies in Table 1. Zero-shot performance on novel classes for models trained on base classes [90, 79, 21] appears in Table 10. Models using novel class images or trained on entire detection datasets are evaluated on two additional datasets (Table 11).

Table 1 shows InstructSAM (esp. with GPT-4o) leading in all tasks with top counting metrics. On novel classes (Table 10), InstructSAM-Qwen achieves superior or competitive $mAP_{nc}$ against specialized models. This underscores InstructSAM's training-free, counting-constrained matching advantage over traditional or fine-tuned approaches.

**Open-ended setting** Table 2 summarizes the results under open-ended setting. InstructSAM consistently achieves equivalent or higher $F_1$-scores than remote sensing-specific methods, including those trained on grounded description tasks [49, 68]. Notably, InstructSAM surpasses LAE-Label [56] by leveraging a global view of the image to accurately predict object categories. While the absence of class-specific instructions in this setting limits further gains, InstructSAM still demonstrates robust performance (Figure 4).

Table 2: Performance comparison on EarthInstruct under open-ended setting.

| Method | NWPU-VHR-10 | | | DIOR | | |
|---|---|---|---|---|---|---|
| | Cnt $F_1$ | Box $F_1$ | Mask $F_1$ | Cnt $F_1$ | Box $F_1$ | Mask $F_1$ |
| Qwen2.5-VL [2] | 48.6 | **32.0** | - | 36.6 | 21.7 | - |
| GPT-4o [24] + OWL [51] | 32.6 | 24.0 | - | 30.6 | 21.0 | - |
| SkySenseGPT [49] | 34.9 | 1.7 | - | 39.2 | 6.5 | - |
| EarthDial [71] | 30.0 | 5.6 | - | 47.6 | 22.8 | - |
| LAE-Label [56] | 46.2 | 27.3 | 24.1 | 23.3 | 11.5 | 10.4 |
| GeoPixel [68] | 40.8 | 29.9 | 29.1 | 21.4 | 13.8 | 14.7 |
| InstructSAM-Qwen | 55.2 | 29.6 | 28.5 | 33.2 | 14.3 | 14.4 |
| InstructSAM-GPT4o | **57.4** | 31.3 | **29.9** | 47.9 | 22.1 | **21.8** |

**Open-subclass setting** Table 3 shows that InstructSAM outperforms or matches Qwen2.5-VL across both parent classes. When prompted with categories identified by GPT-4o, OWLv2 performs strongly for "means of transport" but struggles with "sports fields", likely due to the prevalence of transport-related categories in natural image datasets. These findings are consistent with the open-vocabulary results, where generic detectors, such as Grounding DINO and OWL, struggle with remote sensing categories except for airplane, vehicle, and ship.

Table 3: Performance comparison on EarthInstruct under open-subclass setting. 'S' denotes the parent category "sports field", and 'T' denotes "means of transport".

| Method | NWPU-VHR-10 | | | | | | DIOR | | | | | |
|---|---|---|---|---|---|---|---|---|---|---|---|---|
| | Cnt $F_1$ | | Box $F_1$ | | Mask $F_1$ | | Cnt $F_1$ | | Box $F_1$ | | Mask $F_1$ | |
| | S | T | S | T | S | T | S | T | S | T | S | T |
| Qwen2.5-VL [2] | 54.1 | 48.9 | 32.4 | 42.2 | - | - | 52.2 | 51.8 | 34.0 | 39.2 | - | - |
| GPT-4o [24] + OWL [51] | 40.3 | **68.0** | 19.8 | **65.9** | - | - | 41.5 | **73.6** | 27.6 | **70.9** | - | - |
| InstructSAM-Qwen | 50.9 | 55.3 | 33.5 | 41.9 | 33.0 | 39.7 | 40.4 | 67.4 | 22.2 | 49.3 | 22.6 | **49.5** |
| InstructSAM-GPT4o | **84.2** | 60.6 | **46.9** | 44.2 | **45.8** | **41.6** | **82.2** | 53.6 | **40.9** | 38.3 | **40.2** | 38.6 |

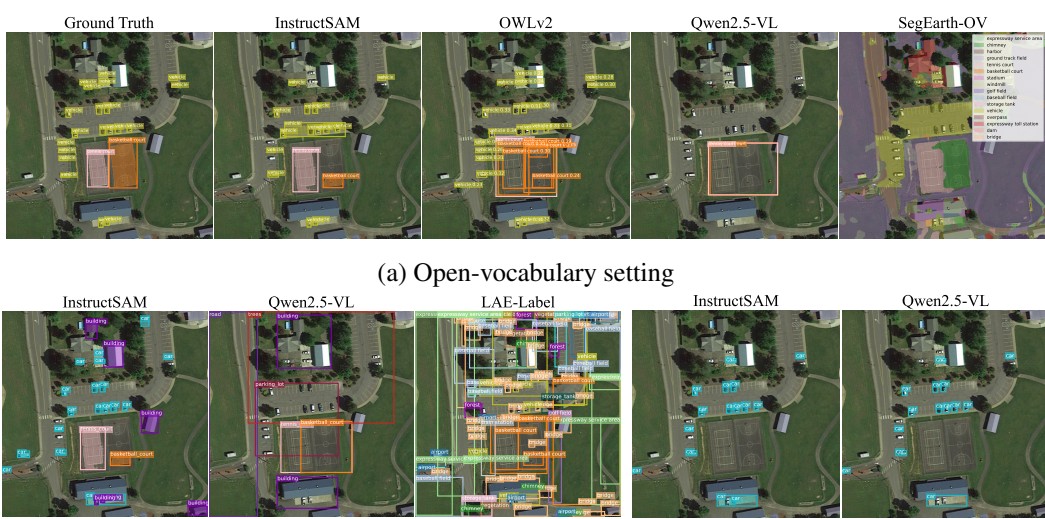

(a) Open-vocabulary setting

(b) Open-ended setting

(c) Open-subclass setting

Figure 4: Qualitative comparison of object detection and segmentation methods on DIOR dataset. InstructSAM consistently outperforms other methods in generating accurate bounding boxes and segmentation masks across various settings. More qualitative results can be found in Appendix F.

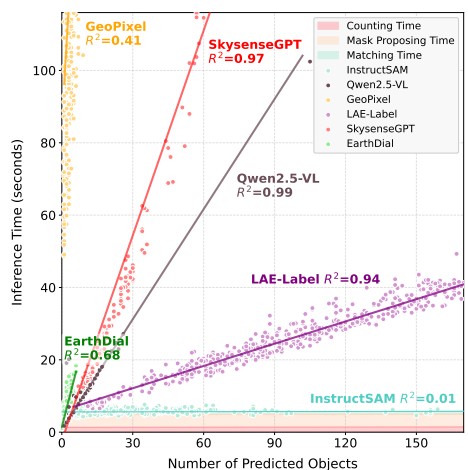

Figure 5: Inference time as a function of bounding box count for InstructSAM-Qwen and baseline methods. Solid lines indicate linear regressions, and scatter points represent individual samples. The shaded regions for InstructSAM illustrate the time composition of different processing steps. Experiments are conducted on an RTX 4090 GPU.

**Inference time analysis**  Figure 5 compares inference times for open-ended methods using 7B LLMs. Mask proposal dominates the runtime of InstructSAM, whereas the PuLP optimizer solves the BIP problem in 0.07 seconds. InstructSAM maintains nearly constant inference time, while other methods scale linearly with the number of objects. Unlike representing bounding boxes as natural language tokens, InstructSAM reduces output tokens by 89% and total inference time by 32% compared to Qwen2.5-VL. This advantage will grow more pronounced as the model size scales up, highlighting the efficiency of our framework.

Table 4: Open vocabulary counting performance (Precision/Recall/$F_1$-score).

| Method | Add Instr | NWPU vehicle | NWPU all | DIOR all |
|---|---|---|---|---|
| Faster-RCNN | ✗ | 15/49/23 | 77/77/73 | 91/74/81 |
| Qwen2.5VL | ✗ | 11/**75**/19 | 80/**72**/72 | 73/**53**/56 |
| Qwen2.5VL | ✓ | **28**/70/**41** | **82**/71/**73** | **83**/51/**59** |
| GPT-4o | ✗ | 11/**80**/20 | 68/79/67 | **87**/65/72 |
| GPT-4o | ✓ | **75**/68/**71** | **83/83/83** | 86/**75/80** |

Table 5: Ablations on model scaling. MP recall is class-agnostic recall of mask proposals.

| LVLM Counter | Mask Proposal | CLIP | MP Recall | Box $F_1$ |
|---|---|---|---|---|
| GPT-4o | SAM2-L | DFN2B-L [13] | 82.4 | 41.3 |
| GPT-4o | SAM2-L | RemoteCLIP-L [43] | 82.4 | 45.7 |
| GPT-4o | SAM2-L | SkyCLIP-L [77] | 82.4 | 45.8 |
| GPT-4o | SAM2-L | SkyCLIP-B [77] | 82.4 | 45.2 |
| GPT-4o | SAM2-S | SkyCLIP-B [77] | 79.1 | 44.1 |
| Qwen2.5VL | SAM2-S | SkyCLIP-B [77] | 79.1 | 40.6 |

## 5.3  Ablation studies

**Prompt design**  Table 4 reveals how additional instructions enhance object counting, particularly for categories with ambiguous or dataset-specific annotation rules. Initially, DIOR-trained Faster-RCNN and LVLM counters exhibit low vehicle precision on NWPU-VHR-10. Explicit annotation rules in instructions significantly boost vehicle precision for Qwen2.5-VL and GPT-4o, and also improve $mF_1$ by 3% and 8%, respectively, on DIOR. Contrary to [92], these results show that capable foundation models with precise, instruction-driven prompts truly empower LVLMs to match or exceed closed-set model performance.

**Model generalization and scaling**  To assess InstructSAM's generalization and scalability, we ablate LVLM counters, mask proposers, and CLIP models for the open-vocabulary detection (OVD) task on NWPU-VHR-10 (Table 5). InstructSAM consistently benefits from CLIP models fine-tuned on remote sensing data [43, 77] over generic CLIP [13], whereby yielding higher Box $F_1$-scores. Performance improves with larger model components, demonstrating the framework's scalability. Notably, even with a smaller SAM2-S and SkyCLIP-B, InstructSAM coupled with Qwen2.5-VL (40.6 Box $F_1$) outperforms direct detection using Qwen2.5-VL alone (36.4 Box $F_1$), underscoring the efficacy of our approach.

**Mask-label matching with counting constraints**  Using fixed thresholds to filter CLIP predictions [10] has inherent limitations. In Figure 6, the green and blue solid curves show InstructSAM without matching, where predictions are filtered by a similarity score (CLIP score) threshold before evaluation. Optimal thresholds vary widely across categories. As a result, a single global value

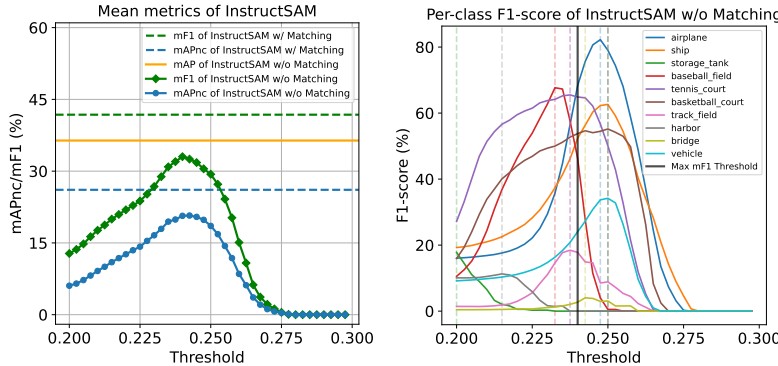

Figure 6: Threshold sensitivity analysis in open-vocabulary setting. Left: Impact on mean metrics. Right: Category-specific $F_1$-scores. Dashed lines indicate optimal thresholds maximizing $mF_1$.

performs poorly, and the results are highly threshold-sensitive, which is consistent with the findings in [36]. In contrast, InstructSAM's counting-constrained matching removes this dependency by dynamically assigning predictions according to the estimated counts, yielding stronger performance in multi-class and open-world settings.

## 5.4 Error analysis of OVD task

Error identification reveals distinct error patterns across methods (Figure 7). OWLv2 primarily suffers from classification errors, while Qwen2.5-VL shows improved classification but struggles with missed detections. InstructSAM-GPT4o benefits from SAM2's localization capabilities, though the background confusion persists due to GeoRSCLIP's scene-focused training, which prioritizes broader contexts over individual objects.

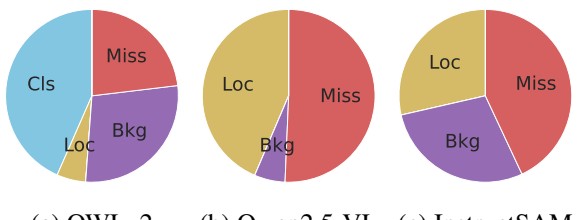

(a) OWLv2     (b) Qwen2.5-VL  (c) InstructSAM

Figure 7: Error distribution of OVD task on NWPU-VHR-10 dataset. Pie charts show proportions of classification (Cls), localization (Loc), background (Bkg), and missed detection (Miss) errors across models.

## 6 Limitations

InstructSAM builds on pre-trained foundation models, and its performance inherits their capabilities and biases. For instance, SAM2 can miss the full extent of objects with intricate geometry (e.g., basketball court in Figure 4). CLIP models finetuned on scene-level remote sensing image–text pairs perform suboptimally when aligning object-level crops with semantic cues. More generally, foundation models such as GPT-4o and SAM are trained primarily on optical imagery and struggle to count or segment objects in synthetic aperture radar (SAR) images.

Future work can mitigate these limitations by (i) incorporating foundation models trained on semantically diverse, multi-modal remote sensing data and (ii) leveraging advanced class-agnostic region proposers [84]. Extending InstructSAM to semantic segmentation would further benefit the earth observation community. Although InstructSAM is only a first step, we aim to use it as an automated labeling engine to annotate at scale, enabling large, semantically rich object-recognition datasets.

## 7 Conclusion

In this paper, we introduce InstructCDS for instruction-driven object counting, detection, and segmentation, along with EarthInstruct, the first benchmark for this task in remote sensing domain. Our training-free InstructSAM framework integrates LVLMs, SAM2, and domain-specific CLIP to handle instruction-oriented scenarios with counting constrained mask-label matching. Experiments demonstrate that InstructSAM outperforms specialized baselines while maintaining near-constant inference time regardless of object counts. As the first approach extending instruction-oriented detection to the broader InstructCDS paradigm, InstructSAM will benefit from advancements in both remote sensing foundation models [42, 85, 86, 67, 69] and general-purpose models [74, 98, 48], paving the way for more scalable, instruction-driven earth observation data analysis.

## Acknowledgement

This work was partly supported by the National Earth Observation Data Center Research Project (Grant No. E43Z18020A), National Natural Science Foundation of China (62506229), and Natural Science Foundation of Shanghai (25ZR1402268).

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

# Technical Appendices

## A   Comparison with related work

Remote sensing imagery captures a vast, diverse range of contexts. Exemplified by the SkyScript dataset [77], curated from Google Earth and OpenStreetMap [19], featuring 29,000 distinct semantic tags. Despite this richness, annotated training datasets for remote sensing object recognition typically span only a few dozen categories (Table 6). This lexical scarcity hinders effective alignment of semantic and visual information compared to generic models, which are typically trained on millions of human-annotated images and near-infinite vocabularies sourced from the Internet. Similarly, instruction-following datasets for remote sensing, such as FIT-RS [49] and GeoPixelD [68], are also built upon manually annotated data, often inheriting these vocabulary-specific limitations.

Regarding model capabilities, remote sensing LVLMs typically fail to follow complex object recognition instructions as illustrated in Figure 8. Nevertheless, SkysenseGPT [49] and GeoPixel [68], trained on scene graph generation or pixel-grounded conversation generation tasks, deliver comprehensive object detection or segmentation outputs, effectively serving a similar purpose of open-ended object recognition.

Table 6: Comparison of language-guided object recognition methods. We summarize the model, publication, training data, dense segmentation capability, and support for instruction-oriented object detection (open-vocabulary detection (OVD), open-ended detection (OED), and open-subclass detection (OSD)). Green ✓ indicates support, red ✗ indicates not supported.

| Method | Publication | Object Recognition Training Data | Dense Segmentation | OVD | OED | OSD |
|---|---|---|---|---|---|---|
| ***Remote sensing-specified methods*** | | | | | | |
| DesReg [90] | AAAI'24 | DIOR | ✗ | ✓ | ✗ | ✗ |
| OVA-DETR [79] | arXiv'24 | DIOR+DOTA+xView | ✗ | ✓ | ✗ | ✗ |
| CASTDet [38] | ECCV'24 | DIOR | ✗ | ✓ | ✗ | ✗ |
| LAE-DINO [56] | AAAI'25 | LAE-1M | ✗ | ✓ | ✗ | ✗ |
| ZORI [21] | AAAI'25 | NWPU/DIOR | ✓ | ✓ | ✗ | ✗ |
| GSNet [88] | AAAI'25 | LandCover-40k | ✓ | ✗ | ✗ | ✗ |
| SegEarth-OV [32] | CVPR'25 | Image-level Training | ✓ | ✗ | ✗ | ✗ |
| SkysenseGPT [49] | arXiv'24 | FIT-RS | ✗ | ✗ | ✓ | ✗ |
| GeoPixel [68] | ICML'25 | GeoPixelD | ✓ | ✗ | ✓ | ✗ |
| ***Generic methods*** | | | | | | |
| GroundingDINO [45] | ECCV'24 | FiveODs+GoldG+Cap4M | ✗ | ✓ | ✗ | ✗ |
| OWLv2 [51] | NIPS'22 | WebLI2B | ✗ | ✓ | ✗ | ✗ |
| Florence-2 [83] | CVPR'24 | FLA-5B | ✗ | ✓ | ✓ | ✗ |
| Qwen2.5-VL [2] | arXiv'25 | Various sources | ✗ | ✓ | ✓ | ✓ |
| InsDetCLIP [60] | ICLR'24 | Object365 | ✗ | ✓ | ✓ | ✓ |
| VL-SAM [41] | NeurIPS'24 | Training-free | ✓ | ✗ | ✓ | ✗ |
| InstructSAM | Ours | Training-free | ✓ | ✓ | ✓ | ✓ |

## B   Benchmark details

The details of the datasets comprising EarthInstruct are as follows:

- **NWPU-VHR-10** [9] is a 10-class dataset for very-high-resolution (VHR) remote sensing object detection. It originally comprises 650 positive images (each containing at least one target object) and 150 negative images (containing no target objects). For constructing EarthInstruct, we utilize the 650 positive images. Instance-level mask annotations were subsequently provided by [72].

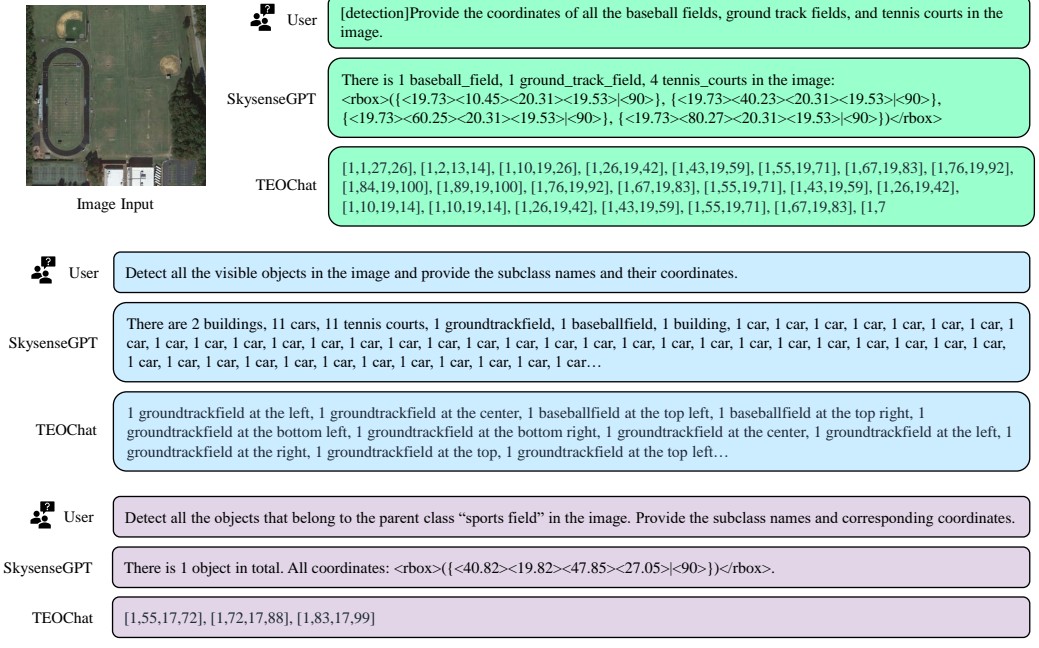

Figure 8: Examples illustrating that SkysenseGPT [49] and TEOChat [25] fail to produce meaningful responses for open-vocabulary, open-ended, and open-subclass prompts. Their responses either lack category name outputs or exhibit looped generation.

- **DIOR** [34] is a large-scale benchmark for object detection in optical remote sensing images, encompassing 20 object categories. The dataset is split into 5,862 training images, 5,863 validation images, and 11,725 test images. For the EarthInstruct benchmark, we utilize the validation set of DIOR. Mask annotations are provided by the SAMRS [75] dataset.

Table 7 provides a summary of these datasets, including their spectral bands, spatial resolution ranges, and the specific category splits within EarthInstruct.

Table 8 compares EarthInstruct with other language-guided object recognition datasets and benchmarks. Most current benchmarks [60, 46, 37] generate prompts using LLMs or templates based on specific objects in the image, resulting in few negative samples. In addition, referring segmenta-

Table 7: Details of the datasets used in EarthInstruct, including spectral bands, resolution, and category splits for base, novel, and open-subclass (sports field, means of transport) settings. "Res" denotes spatial resolution. "NWPU" stands for NWPU-VHR-10.

| Dataset | Band | Res | Base Class | Novel Class | Sports Field | Means of Transports |
|---|---|---|---|---|---|---|
| NWPU | RGB/ Color Infrared | 0.08-2m | airplane, storage_tank, baseball_field, tennis_court, track_field, bridge, vehicle | ship, basketball_ court, harbor | baseball_field, tennis_court, basketball_court, track_field | airplane, ship, vehicle |
| DIOR | RGB | 0.3-30m | airplane, baseball field, bridge, chimney, expressway service area, expressway toll station, dam, golf field, harbor, overpass, ship, stadium, storage tank, tennis court, train station, vehicle | airport, basketball court, ground track field, windmill | baseball field, basketball court, golf field, ground track field, tennis court, stadium | airplane, ship, vehicle |

Table 8: Comparison between EarthInstruct and other language-guided object recognition benchmark datasets. "OV", "OE", "OS" denote open-vocabulary, open-ended, and open-subclass, respectively. Goal-Oriented denotes instructions to detect objects that achieve certain goals. **Content Prior Free** means the benchmark does **not** require image content prior for prompt construction.

| Dataset | Publication | OV Instructions | OE Instructions | OS Instructions | Goal-Oriented Instructions | Content Prior Free | Class Num |
|---|---|---|---|---|---|---|---|
| IOD-Bench [60] | ICLR'24 | ✓ | ✓ | ✓ | ✓ | ✗ | – |
| RRSIS-D [46] | CVPR'24 | ✓ | ✗ | ✗ | ✗ | ✗ | 20 |
| VRSBench [37] | NeurIPS'24 | ✓ | ✗ | ✗ | ✗ | ✗ | 26 |
| FIT-RS (OD) [49] | arXiv'24 | ✓ | ✗ | ✗ | ✗ | ✗ | 48 |
| FIT-RS (SGG) [49] | arXiv'24 | ✗ | ✓ | ✗ | ✗ | ✓ | 48 |
| GeoPixelID [68] | ICML'25 | ✗ | ✓ | ✗ | ✗ | ✓ | 15 |
| EarthReason [33] | arXiv'25 | ✗ | ✗ | ✗ | ✓ | ✗ | 28 |
| EarthInstruct | Ours | ✓ | ✓ | ✓ | ✗ | ✓ | 20 |

tion [46] and visual grounding [37] benchmarks typically support only single-class or single-object queries, restricting efficiency in large-scale applications.

EarthInstruct addresses these issues by supporting open-vocabulary, open-ended, and open-subclass settings, and uses dataset-specific prompts that encode user intent without relying on image content priors. This enables more comprehensive and efficient evaluation for instruction-driven object recognition.

## C  Implementation details

This section provides further details on the implementation of our InstructSAM framework and other baselines. A critical component of leveraging LVLMs for the InstructCDS tasks is the design of effective prompts. We employ structured prompts in JSON format to systematically guide the LVLM counter. This structured approach allows for clear delineation of the model's `Persona`, the specific `Task` to be performed, detailed `Instructions` for execution, the desired `Output format`, and illustrative `Examples`. The `Instructions` field enables the incorporation of dataset-specific annotation rules and disambiguation of category definitions, which are crucial for accurate interpretation the user intension, as discussed in Section 3.1. The subsequent subsections detail the specific prompts used for each setting within the EarthInstruct benchmark.

Here we further clarify how the categories that might have bewildered the model. For example, an "overpass" is defined as "a bridge, road, railway or similar structure that is over another road or railway[1]", while a "bridge" is "a structure built to span a physical obstacle (such as a body of water, valley, road, or railway) without blocking the path underneath[2]." It is possible that the LVLM counter will recognize overpasses as bridges since overpass is a subset of bridge. Adding clarification such as "bridge in this dataset refers to a structure spanning a body of water" to the `Instructions` field will guild the LVLM to count accurately.

### C.1  Prompts in open-vocabulary setting

- Prompt for NWPU-VHR-10 dataset:

```
{
    "Persona": "You are an advanced AI model capable of understanding and
    analyzing aerial images.",
    "Task": "Given an input satellite imagery, count the number of objects from
    specific categories. Provide the results in JSON format where the keys are
    the category names and the values are the corresponding counts.",
    "Instructions": [
```

---

[1]https://en.wikipedia.org/wiki/Overpass
[2]https://en.wikipedia.org/wiki/Bridge

```
        "The 10 categories in the dataset are: ['airplane', 'ship',
    'storage_tank', 'baseball_field', 'tennis_court', 'basketball_court',
    'track_field', 'harbor', 'bridge', 'vehicle']",
        "The spatial resolution of the imagery in the dataset ranges from 0.08 m
    to 2 m.",
        "Do not count ships or vehicles that are hard to annotate in the
    relatively low-resolution images as they are not annotated due to the small
    size.",
        "Harbor is defined as a pier to dock ships. If multiple harbors are
    visible in the image, count each distinct pier separately."
    ],
    "Output format": "{\"category1\": count1, \"category2\": count2, ... }",
    "Examples": [
        { "airplane": 2, "ship": 0, "storage_tank": 3, "baseball_field": 1,
    "tennis_court": 0, "basketball_court": 0, "track_field": 0, "harbor": 6,
    "bridge": 0, "vehicle": 0 },
        { "airplane": 5, "ship": 2, "storage_tank": 0, "baseball_field": 0,
    "tennis_court": 1, "basketball_court": 0, "track_field": 0, "harbor": 0,
    "bridge": 1, "vehicle": 10 }
    ]
}
```

- Prompt for DIOR dataset:

```
{
    "Persona": "You are an advanced AI model capable of understanding and
    analyzing remote sensing images."
    "Task": "Given an input satellite imagery, count the number of objects from
    specific categories. Provide the results in JSON format where the keys are
    the category names and the values are the corresponding counts.",
    "Instructions": [
        "The 20 categories in the dataset are: ['airplane', 'airport', 'baseball
    field', 'basketball court', 'bridge', 'chimney', 'expressway service area',
    'expressway toll station', 'dam', 'golf field', 'ground track field',
    'harbor', 'overpass', 'ship', 'stadium', 'storage tank', 'tennis court',
    'train station', 'vehicle', 'windmill']",
        "The spatial resolution of the images is 0.3m-30m.",
        "Airport is a large area of land where aircraft can take off and land. It
    includes runways and other facilities. Do not count airport if the it is not
    compeletely visible in the image.",
        "Harbor is defined as a pier to dock ships. If multiple harbors are
    visible in the image, count each distinct pier separately.",
        "Expressway toll station is a toll booth at the entrance of the
    expressway and spans the road.",
        "Expressway service area is a rest area along an expressway. If it exsits
    on the both sides of the expressway, count them separately.",
        "Overpass is a road crossing over another road. Bridge is a road spanning
    a river. Distinguish them carefully.",
        "If the overpass or bridge is composed of parallel, separate sections
    (for example, different lanes or directions of traffic), each section should
    be counted individually.",
        "Count every ship and vehicle carefully, even the resolution is low and
    the objects are small and dense.",
        "If none of the objects among the categories is visible, output a JSON
    object with all categories set to 0"
    ],
    "Output format": "{ "category1": count1, "category2": count2, ... }",
    "Examples": [
        "{ "airplane": 2, "airport": 0, "baseball field": 0, "basketball court":
    0, "bridge": 1, ... }",
        "{ "airplane": 0, "airport": 0, "baseball field": 2, "basketball court":
    6, "bridge": 0, ... }"
    ]
}
```

## C.2 Prompts in open-ended setting

• Prompt for NWPU-VHR-10 dataset:

```
{
    "Persona": "You are an advanced AI model capable of understanding and
    analyzing remote sensing images.",
    "Task": "Given an input satellite imagery, count the number of all the
    visible remote sensing objects. Provide the results in JSON format where the
    keys are the category names and the values are the corresponding counts.",
    "Instructions": [
        "The spatial resolution of the imagery in the dataset ranges from 0.08 m
    to 2 m.",
        "Do not count ships or vehicles that are too samll and are hard to
    annotate in the relatively low-resolution images.",
        "Only count objects that are clearly visible in the imagery. If a
    category is not visible, do not include it in the output."
    ],
    "Output format": "{ "category1": count1, "category2": count2, ... }",
    "Answer": [
        "Ensure the category names are in singular form",
        "Provide the counts as integers."
    ]
}
```

• Prompt for DIOR dataset:

```
{
    "Persona": "You are an advanced AI model capable of understanding and
    analyzing remote sensing images.",
    "Task": "Given an input satellite imagery, count the number of all the
    visible remote sensing objects or scenes. Provide the results in JSON format
    where the keys are the category names and the values are the corresponding
    counts.",
    "Instructions": [
        "The spatial resolution of the imagery in the dataset ranges from 0.3 m
    to 30 m.",
        "If the resolution is too limited or the scene is too dense to accurately
    count certain objects, exclude those objects from the results.",
        "Only count objects that are clearly visible in the imagery."
    ],
    "Output format": "{ "category1": count1, "category2": count2, ... }",
    "Answer": [
        "Ensure the category names are in singular form",
        "Provide the counts as integers."
    ]
}
```

## C.3 Prompts in open-subclass setting

• Prompt for counting "sports field" in NWPU-VHR-10 dataset:

```
{
    "Persona": "You are an advanced AI model capable of understanding and
    analyzing remote sensing images."
    "Task": "Given an input satellite imagery, count the number of objects that
    belong to the parent category **sports field**. Provide the results in JSON
    format where the keys are the names of the subcategories and the values are
    the corresponding counts.",
    "Instructions": [
        "The spatial resolution of the images is 0.08m-2m.",
        "Do not count objects that are hard to annotate in the relatively
    low-resolution images as they are not annotated due to the small size.",
        "If none of the objects belong to the parent category is visible, output
    a empty JSON object like \{ \}"
```

```
    ],
    "Output format": "{ "subcategory1": count1, "subcategory2": count2, ... }",
    "Answer": [
        "Ensure the category names are in singular form",
        "Provide the counts as integers."
    ]
}
```

- Prompt for counting "means of transport" in NWPU-VHR-10 dataset:

```
{
    "Persona": "You are an advanced AI model capable of understanding and
    analyzing remote sensing images."
    "Task": "Given an input satellite imagery, count the number of objects that
    belong to the parent category **means of transport**. Provide the results in
    JSON format where the keys are the names of the subcategories and the values
    are the corresponding counts.",
    "Instructions": [
        "The spatial resolution of the imagery in the dataset ranges from 0.08 m
    to 2 m.",
        "Do not count boats or land vehicles that are hard to annotate in the
    relatively low-resolution images as they are not annotated due to the small
    size.",
        "If none of the objects belong to the parent category is visible, output
    a empty JSON object like \{ \}"
    ],
    "Output format": "{ "subcategory1": count1, "subcategory2": count2, ... }",
    "Answer": [
        "Ensure the category names are in singular form",
        "Provide the counts as integers."
    ]
}
```

- Prompt for counting "sports field" in DIOR dataset:

```
{
    "Persona": "You are an advanced AI model capable of understanding and
    analyzing remote sensing images."
    "Task": "Given an input satellite imagery, count the number of visible
    objects that belong to the parent category **sports field**. Provide the
    results in JSON format where the keys are the names of the subcategories and
    the values are the corresponding counts.",
    "Instructions": [
        "The spatial resolution of the images is 0.3m-30m.",
        "Return only the categories and counts that meet the visibility and
    resolution criteria. If none of the objects belong to the parent category is
    visible, output a empty JSON object like \{ \}".
    ],
    "Output format": "{ "subcategory1": count1, "subcategory2": count2, ... }",
    "Answer": [
        "Ensure the category names are in singular form",
        "Provide the counts as integers."
    ]
}
```

- Prompt for counting "means of transport" in DIOR dataset:

```
{
    "Persona": "You are an advanced AI model capable of understanding and
    analyzing remote sensing images."
    "Task": "Given an input satellite imagery, count the number of visible
    objects that belong to the parent category **means of transport**. Provide
    the results in JSON format where the keys are the names of the subcategories
    and the values are the corresponding counts.",
    "Instructions": [
        "The spatial resolution of the images is 0.3m-30m.",
```

```
        "Return only the categories and counts that meet the visibility and
    resolution criteria. If none of the objects belong to the parent category is
    visible, output a empty JSON object like \{ \}"
    ],
    "Output format": "{ "subcategory1": count1, "subcategory2": count2, ... }",
    "Answer": [
        "Ensure the category names are in singular form",
        "Provide the counts as integers."
    ]
}
```

## C.4  Hyperparameters of InstructSAM

**Mask proposer**   As described in Section 4.2, class-agnostic mask proposals are generated using SAM2-hiera-large [63] in its automatic mask generation mode. The model was configured with the parameters detailed in Table 9 across both NWPU-VHR-10 and DIOR. These settings balance the proposal quality and computational efficiency, ensuring a dense set of high-quality proposals, especially for small objects.

Table 9: Hyperparameters for SAM2.

| Parameter | Value |
|---|---|
| pred_iou_thresh | 0.75 |
| stability_score_thresh | 0.75 |
| points_per_side | 24 |
| crop_n_layers | 1 |
| box_nms_thresh | 0.5 |

**LVLM counter**   For Qwen2.5-VL and GPT-4o, the temperature is set at 0.01 and top_p is set at 1 to reduce randomness.

## C.5  Implementations of baseline methods

To ensure rigorous and fair comparisons against InstructSAM, baseline methods are implemented or adapted as detailed below. Unless otherwise specified, publicly available pre-trained models and official codebases are utilized, adhering to their recommended configurations.

**Open-Vocabulary semantic segmentation methods**   For SegEarth-OV [32] and GSNet [88], we follow their prescribed inference procedures. This involves providing the target categories along with an explicit "background" category as input. To derive instance-level proposals from the pixel-level segmentation maps produced by these models, we apply a standard connected component labeling algorithm to the per-class binary masks. This allows evaluation using instance-based metrics (Mask $F_1$).

**Open-ended methods**

- **SkysenseGPT** [49]: We utilize the prompt "[grounding]Analyze and describe every detail you can identify in the image," which is a representative of its training data, and set the maximum output token limit to 5000.
- **GeoPixel** [68]: Following the illustrative examples provided in their publication, we employ the prompt "Can you give a thorough description of this image, including interleaved segmentation masks to highlight key objects?"
- **LAE-Label** [56]: For mask proposing, we use SAM2-hiera-large with the same hyperparameters as InstructSAM (see Appendix C.4). For label generation, we replace LAE-Label's original InterVL2-8B and reasoning prompt with the newer Qwen2.5-VL-7B, guided by a structured prompt for open-ended category identification without explicit reasoning. This change of prompt improves $mF_1$ by 4.3% on NWPU-VHR-10 and reduces inference time by 92%. The prompt we use is as follows:

```
{
    "Persona": "You are an advanced AI model capable of understanding and
    analyzing remote sensing images."
    "Task": "Given an input satellite imagery, identify the most likely object
    class visible in the image. Provide the result as a single class name."
    "Instructions": [
        "Consider this is a region of interest cropped from a larger remote
    sensing image.",
        "Focus on identifying the main object class visible in the cropped
    region.",
        "Be specific with your answer, using a single class name."],
    "Output format": "\"class_name\"",
    "Answer": ["Provide only the class name in quotes, without additional
    explanations. If it is not recognized, output \"Unrecognized\" "]
}
```

**Qwen2.5-VL**  Given Qwen2.5-VL's [2] extensive pre-training on object detection tasks, we request Qwen2.5-VL to generate a prompt template to perform IOD, and specify the output format following the examples in their paper. Taking DIOR dataset as an example, the prompts for each task are as follows.

- Open-vocabulary detection:

```
**Prompt:**
I need assistance with performing an remote sensing object detection task using
    the provided category names: ['airplane', 'airport', 'baseball field',
    'basketball court', 'bridge', 'chimney', 'expressway service area',
    'expressway toll station', 'dam', 'golf field', 'ground track field',
    'harbor', 'overpass', 'ship', 'stadium', 'storage tank', 'tennis court',
    'train station', 'vehicle', 'windmill'].
Please provide the results in JSON format, where each object is represented as
    follows:
- Each object should include a label (category name) and its corresponding
    bounding box coordinates (in the format [x1, y1, x2, y2]).
**Example Output:**
[
    {"label": "category_name", "bbox_2d": [x1, y1, x2, y2]},
    {"label": "category_name", "bbox_2d": [x1, y1, x2, y2]},
    ...
]
```

- Open-ended detection:

```
**Prompt:**
Please detect all the visible objects in the satellite image and provide the
    results in JSON format, where each object is represented as follows:
- Each object should include a label (category name) in singular form and its
    corresponding bounding box coordinates (in the format [x1, y1, x2, y2]).
**Example Output:**

[
    {"label": "category1", "bbox_2d": [x1, y1, x2, y2]},
    {"label": "category2", "bbox_2d": [x1, y1, x2, y2]},
    ...
]
```

- Open-subclass detection (e.g., for "sports field"):

```
**Prompt:**
Please detect all the visible objects in the satellite image that belong to the
    parent category **sports field** and provide the results in JSON format,
    where each object is represented as follows:
```

```
- Each object should include a label (subcategory name) in singular form and its
    corresponding bounding box coordinates (in the format [x1, y1, x2, y2]).
- If none of the objects belong to the parent category is visible, output a empty
    empty list like [].
**Example Output:**
[
    {"label": "subcategory1", "bbox_2d": [x1, y1, x2, y2]},
    {"label": "subcategory2", "bbox_2d": [x1, y1, x2, y2]},
    ...
]
```

While incorporating detailed dataset-specific instructions into prompts enhances Qwen2.5-VL's performance for object counting, such elaborations does not benefit direct detection. In fact, such instructions reduce $mF_1$ by 1% on NWPU-VHR-10 and have little effect on DIOR. The main paper reports open-vocabulary detection results without these instructions.

## D Additional experiments

### D.1 Comparison with zero-shot remote sensing object recognition methods on EarthInstruct

Direct comparison with some zero-shot remote sensing detection and segmentation methods (e.g., DesReg [90], OVA-DETR [79], ZoRI [21]) is difficult due to differing dataset splits, evaluation metrics, and limited reproducibility. Table 10 shows their reported mAP scores alongside InstructSAM's $mAP_{nc}$ on novel classes. Notably, InstructSAM-Qwen matches or outperforms these methods on their respective benchmarks (e.g., ZoRI on NWPU-VHR-10, OVA-DETR on DIOR). While these baselines use confidence-based AP, InstructSAM's confidence-free APnc results underscore its robustness.

Table 10: Comparison with zero-shot remote sensing object detection and segmentation methods on **novel** classes. '-' means data missing due to limited reproducibility. $\emptyset$ indicates model lacking the segmentation ability. APnc is reported for InstructSAM, while values in gray for other methods are AP reported in their original papers.

| Method | NWPU-VHR-10 | | DIOR val | | DIOR test | |
|---|---|---|---|---|---|---|
| | Box AP | Mask AP | Box AP | Mask AP | Box AP | Mask AP |
| DesReg [90] | - | $\emptyset$ | - | $\emptyset$ | 7.9 | $\emptyset$ |
| OVA-DETR [79] | - | $\emptyset$ | 7.1 | $\emptyset$ | - | $\emptyset$ |
| ZoRI [21] | - | 12.3 | - | - | - | 8.5 |
| InstructSAM-Qwen | 24.6 | 24.1 | 7.6 | 6.3 | 4.9 | 4.3 |
| InstructSAM-GPT4o | 26.8 | 26.5 | 11.2 | 8.9 | 6.6 | 5.4 |

### D.2 Comparison with open-vocabulary detection methods on out-of-distribution datasets

To compare with open-vocabulary methods that include DIOR into training, we compare the zero-shot performance on two out-of-distribution (OOD) datasets:

- **xBD** [18] is a large-scale building damage assessment dataset with spatial resolution below 0.8m. The test set includes 933 pre- and post-disaster image pairs with instance-level building masks. We use pre-disaster images to evaluate object counting and detection with the single category "building".
- **Aerial Maritime Drone Large** [30] is a drone-based object detection dataset with 74 aerial maritime images and 1,151 bounding boxes. Categories include "docks", "boats", "lifts", "jetskis", and "cars". The entire dataset is used for zero-shot evaluation.

Table 11 shows that LAE-DINO, trained on LAE-1M (including "building"), achieves the highest detection $F_1$-core (50.6) on xBD. On Aerial Maritime Drone, however, LAE-DINO detects only boats, even struggling with docks and cars despite semantic similarity to harbor and vehicle categories in training. Moreover, the confidence-based detectors (OWL, CASTDet, LAE-DINO) require dataset-

Table 11: Comparison with open-vocabulary detection methods on xBD and Aerial Maritime Drone Large datasets.

| Method | xBD | | | Aerial Maritime Drone Large | | |
|---|---|---|---|---|---|---|
| | Best Thr | Cnt mF$_1$ | Det mF$_1$ | Best Thr | Cnt mF$_1$ | Det mF$_1$ |
| OWL [51] | 0.02 | 53.0 | 31.5 | 0.26 | 28.3 | 21.5 |
| CASTDet [38] | 0.00 | 0.0 | 0.0 | 0.40 | 13.3 | 10.8 |
| LAE-DINO [56] | 0.08 | 79.6 | 50.6 | 0.30 | 13.0 | 12.5 |
| InstructSAM-GPT4o | - | 65.9 | 37.2 | - | 44.5 | 22.5 |

specific thresholds to optimize zero-shot results, highlighting InstructSAM's advantage in eliminating threshold tuning.

### D.3 Selection of prompt format

We conduct five independent experiments evaluating Qwen2.5-VL and GPT-4o's open-vocabulary counting ability on NWPU-VHR-10 using Markdown and JSON prompts. Table 12 shows the mean accuracies and standard deviations. Qwen2.5-VL exhibits only a 1.2% difference in mean accuracy between Markdown and JSON, with zero standard deviation for both. GPT-4o shows slight variability: 82.38% ± 0.94 (Markdown) and 82.68% ± 0.39 (JSON). These results indicate that these LVLM counters maintain relatively stable performance across different prompt formats.

Table 12: Mean accuracy and 1-sigma standard deviation for different prompt formats and models.

| Model | Prompt Format | Mean Accuracy (%) | 1-Sigma (Stand Deviation) |
|---|---|---|---|
| Qwen2.5-VL | Markdown | 72.0 | 0.00 |
| Qwen2.5-VL | JSON | 73.2 | 0.00 |
| GPT-4o | Markdown | 82.4 | 0.94 |
| GPT-4o | JSON | 82.7 | 0.39 |

## E   Broader impacts

InstructSAM demonstrates a strong ability to recognize objects in remote sensing imagery with versatile instructions. It accelerates large-scale mapping applications, further supporting public agencies and humanitarian organizations in critical areas such as poverty mapping, disaster response. Its training-free paradigm and near-constant inference speed lower computational costs, decreasing carbon emissions and broadening access to remote sensing applications in resource-limited settings.

InstructSAM's capabilities also raise privacy concerns as the spatial resolution of satellite imagery is growing higher. Hallucinations or misclassifications, stemming from biases in pre-trained models may produce unreliable outputs, require user verification to ensure accuracy.

## F   Qualitative results

This section presents qualitative results of InstructSAM across open-vocabulary (Figure 9), open-ended (Figure 10), and open-subclass settings (Figure 11). Additionally, we showcase its versatility in handling instructions to achieve certain goals in Figure 12 and its generalization to natural images in Figure 13.

## F.1 Qualitative results in open-vocabulary setting

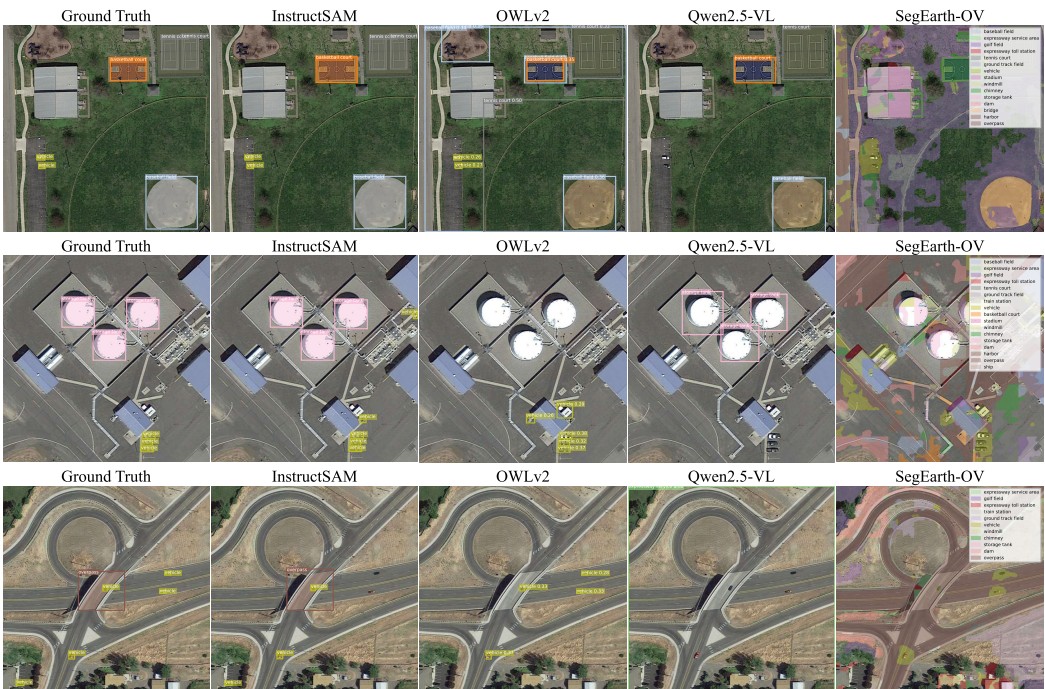

Figure 9: Qualitative results in open-vocabulary setting. While OWLv2 struggles to distinguish remote sensing objects beyond vehicles, and SegEarth-OV fails to separate foreground objects from the background, InstructSAM demonstrates superior performance in segmenting remote sensing objects.

## F.2   Qualitative results in open-ended setting

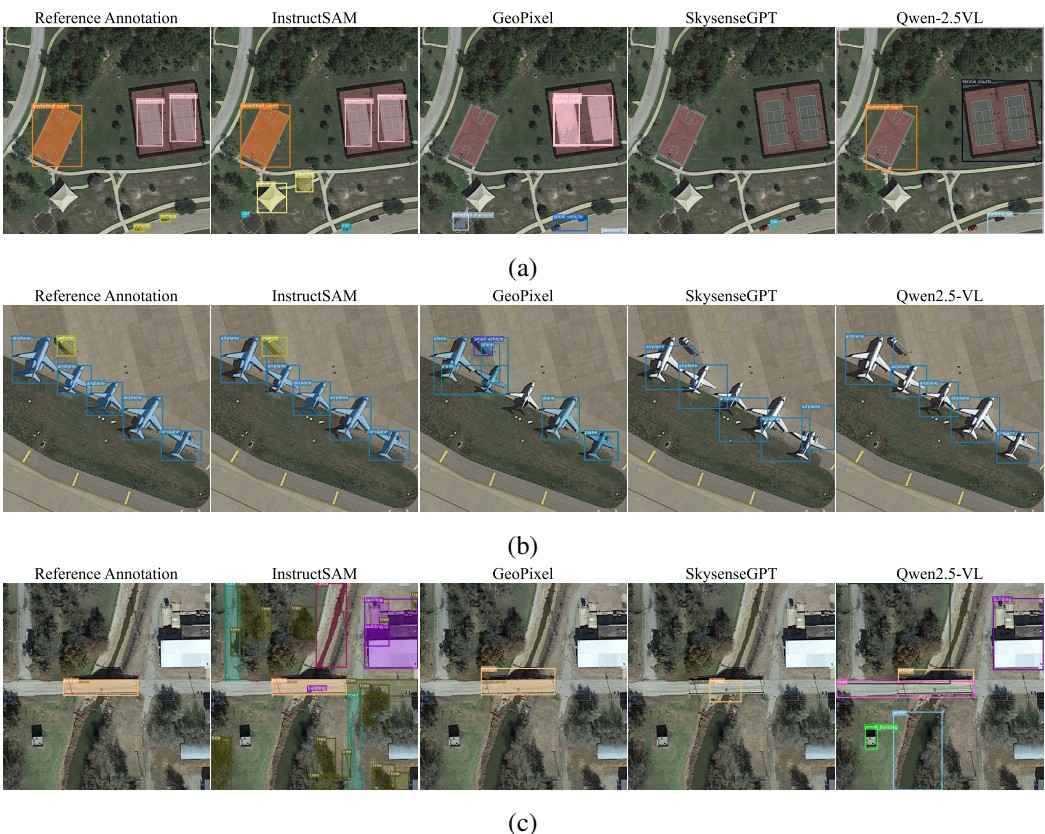

(a)

(b)

(c)

Figure 10: Qualitative results in open-ended setting. Unlike GeoPixel and SkysenseGPT, which fail to detect classes outside their training set, InstructSAM demonstrates its ability to recognize diverse objects (e.g., pavilion in (a), tree in (c)) and provides more accurate bounding boxes and less fragmented masks.

### F.3 Qualitative results in open-subclass setting

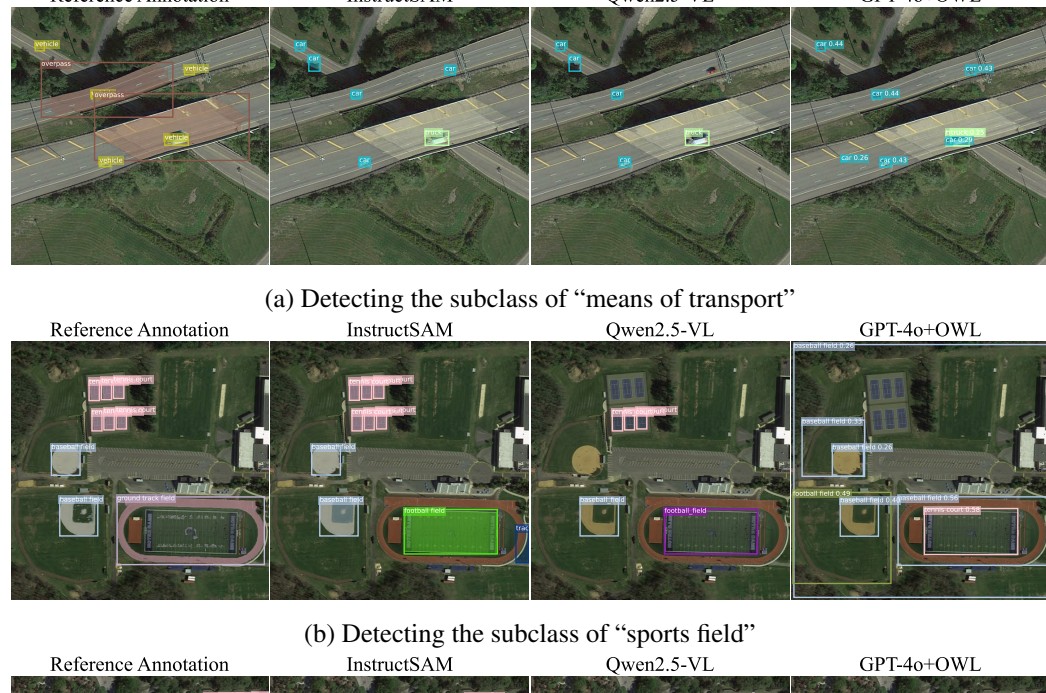

(a) Detecting the subclass of "means of transport"

(b) Detecting the subclass of "sports field"

(c) Detecting the subclass of "sports field"

Figure 11: Qualitative results in open-subclass setting. InstructSAM effectively identifies objects within parent categories. In contrast, Qwen2.5-VL struggles with dense objects, and OWLv2 faces challenges in classifying sports fields from a bird's-eye view.

## F.4 Qualitative results on other instructions

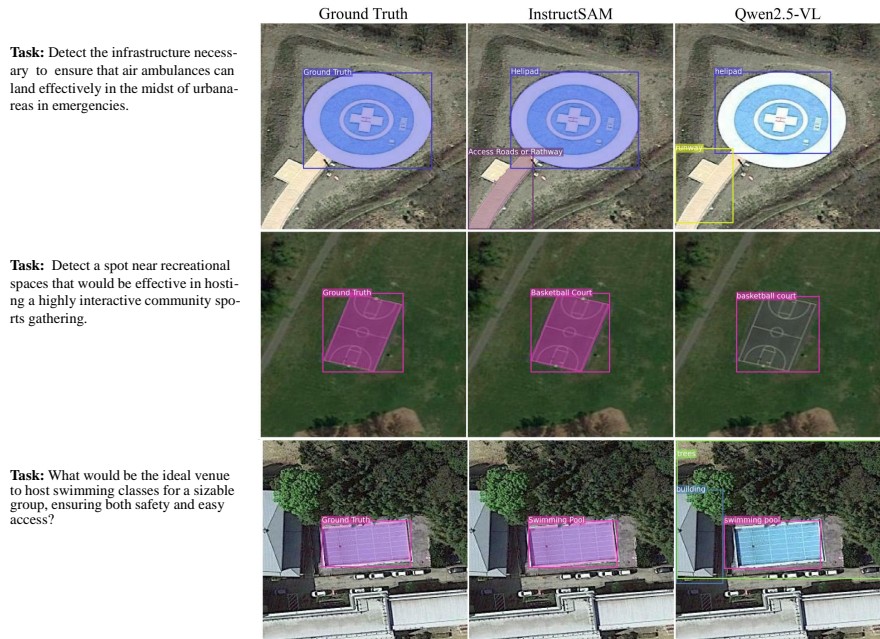

Figure 12: Qualitative results on following versatile instructions. Samples are from EarthReason [33] dataset. InstructSAM successfully recognizes objects based on implicit cues.

## F.5 Qualitative results in natural images

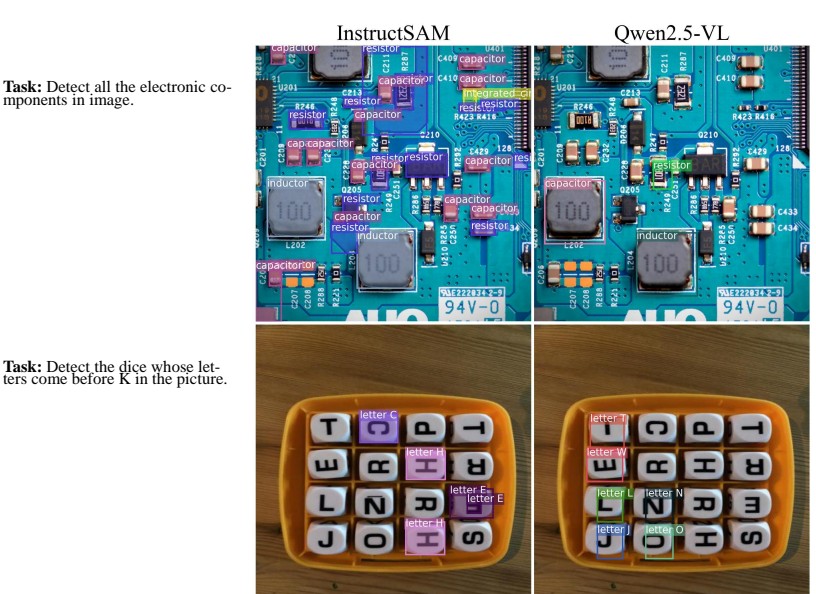

Figure 13: Qualitative results on natural images. When equipped with generic CLIP models (DFN2B-CLIP [13]), InstructSAM effectively recognizes objects in natural images.

