# OpenReview forum: "InstructSAM: A Training-free Framework for Instruction-Oriented Remote Sensing Object Recognition"
_NeurIPS.cc/2025/Conference — NeurIPS 2025 poster_

### Official Review · Reviewer_te26 · 2025-06-25

**Clarity:** 2
**Significance:** 2
**Originality:** 3
**Rating:** 5
**Confidence:** 4

**Summary:**

The paper introduces InstructSAM, a training-free pipeline that turns a natural-language instruction into object counts, masks and labels in remote-sensing images. It pairs a vision-language model for counting with SAM2 for mask proposals. The authors also release EarthInstruct, a small benchmark to test this “instruction counting-detection-segmentation” task.

**Questions:**

1.	RS domain: Why did you focus your experiments on RS when you are using models from the natural domain? Why not evaluating on natural images as well?
2.	Broader evaluation:  As your paper focus on RS, additional experiments with other datasets would improve the paper, specifically, low-res datasets (e.g. S-2) or dataset with more diverse class types (e.g. from land-use or agriculture).
3.	Out-of-distrubution: You are testing InstructSAM on two further datasets in the appendix. Why are they called “out-of-distribution”? You are using generic models (SAM2 and CLIP) that are most likely trained on these categories as well.
4.	Task novelty claim: As you clarify in the related work, InstructCDS does not fundamentally differs from existing instruction-oriented tasks and your claim “new suite of tasks” might be a bit overstated.

**Ethical Concerns:**

["NO or VERY MINOR ethics concerns only"]

**Final Justification:**

Technically solid paper, main concerns regarding dataset diversity and task novelty were adressed during the rebuttal.

**Limitations:**

Yes

**Paper Formatting Concerns:**

Formating of Section headers

**Quality:**

3

**Strengths And Weaknesses:**

Strengths
- Training-free approach.
- Works with different VL models and reduces runtime compared to other models.
- The BIP problem removes hand-tuned thresholds for label assignment.
- Covers multiple tasks (conting, detection, segmentation) in multiple settings.
- Very good results compared to benchmark models.

Weaknesses
- Tested only on two datasets covering two classes categories. Only high-res data is evaluated most public RS data is medium-res (e.g. Sentinel 2 with 10m).
- Task novelty overlaps earlier instruction-based RS works; “first” claim overstated.
- Section headers do not follow the style guidelines.
- Real world applicability is limited due to very specific architecture design and limited generalization experiments.

---

> ### Author Rebuttal · Authors · 2025-07-31
>
> **Weaknesses**
>
> > Tested only on two datasets covering two classes categories. Only high-res data is evaluated most public RS data is medium-res (e.g. Sentinel 2 with 10m).
>
> Thank you for this observation. Most object recognition datasets in remote sensing are with high-resolution. DIOR was chosen for its wide GSD range (0.3m to 30m), which includes medium-resolution images. Many categories, such as "ship," "airport," "expressway service area," and "ground track field," frequently occur in medium-resolution images (as shown in Figure 2(d)). Additionally, EarthInstruct is a multimodal (NIR+RGB) and multi-resolution (0.08m–30m) benchmark, making it representative of diverse remote sensing scenarios.
>
> > Task novelty overlaps earlier instruction-based RS works; “first” claim overstated.
>
> We appreciate your feedback. While there are overlaps with earlier works like EarthDial [1] and TEOChat, we believe InstructCDS and EarthInstruct introduce significant novelties:
>
> * **Broader instruction scope:** Existing works are limited to counting or detecting **one object type per query**. In contrast, EarthInstruct benchmark is the first benchmark that supports **open-vocabulary**,**​ open-ended**, and **open-subclass** settings in EO, enabling much more complex and diverse instructions.
> * **Structured outputs:** Earlier works fail to generate structured outputs for multi-class tasks. InstructSAM provides structured, interpretable results, aligning with complex user-defined instructions.
>
> > Section headers do not follow the style guidelines.
>
> Thank you for pointing this out. We will ensure all section headers follow the style guidelines, with only the first word and proper nouns capitalized.
>
> > Real world applicability is limited due to very specific architecture design and limited generalization experiments.
>
> We acknowledge this concern and have added model generalization and scaling experiments to address it. InstructSAM demonstrates good generalization across various open LVLMs, as shown below.
>
> | Model                 | Counting mAcc | Counting mRecall | Counting mF1 |
> | ----------------------- | --------------- | ------------------ | -------------- |
> | Gemma3-12B            | 73.2          | 76.4             | 72.8         |
> | Mistral-Samll-3.2-24B | 84.5          | 65.6             | 70.8         |
> | Qwen2.5-VL-7B         | 81.9          | 70.5             | 73.2         |
> | Qwen2.5-VL-72B        | 81.2          | 83.9             | 79.5         |
>
> Additionally, InstructSAM's "out-of-box" design allows it to benefit from any improved open-source or proprietary foundation models, enhancing its applicability. The use of LVLM allows InstructSAM to interpret diverse instructions.
>
> **Questions**
>
> > RS domain: Why did you focus your experiments on RS when you are using models from the natural domain? Why not evaluate natural images as well?
>
> Our focus is on remote sensing (RS) due to its unique challenges, such as the lack of semantically diverse training data. In contrast, the natural domain benefits from abundant datasets like Object365, LVIS, and V3Det, making it relatively straightforward to achieve InstructCDS using models like GPT4o+OWLv2 or by training a generative model such as Ins-DetCLIP [1].
>
> Also, we observed that SAM struggles to segment complex objects (e.g., person, car) as a single entity in natural images. However, this issue is less pronounced in satellite images due to the different texture features with natural images.
>
> That said, we conducted qualitative experiments on natural images, such as electronic components and dice detection (Fig. 13). In these cases, InstructSAM outperformed generic detectors like OWLv2 and Qwen2.5-VL. Quantitative evaluations on natural images remain an interesting direction for future work.
>
> > Broader evaluation: As your paper focus on RS, additional experiments with other datasets would improve the paper, specifically, low-res datasets (e.g. S-2) or dataset with more diverse class types (e.g. from land-use or agriculture).
>
> Thank you for this suggestion. While DIOR already includes medium-resolution images, we agree that additional experiments on low-resolution datasets or those with diverse class types (e.g., land-use or agriculture) would strengthen the paper. Here we demonstrate InstructSAM's superior zero-shot ability in land cover classification on FLAIR dataset compared to the recent open-vocabulary segmentation method. Broader evaluation is an exciting direction for future work.
>
> | Model                 | building | pervious surface | impervious surface | bare soil | water | coniferous | deciduous | brushwood | vineyard | herbaceous vegetation | agricultural land | plowed land | swimming pool | mIoU |
> | ----------------------- | ---------- | ------------------ | -------------------- | ----------- | ------- | ------------ | ----------- | ----------- | ---------- | ----------------------- | ------------------- | ------------- | --------------- | ------ |
> | SegEarth-OV       | 47.6     | 6.5              | 0.1                | 13.7      | 49.6  | 10.6       | 3.1       | 4.2       | 34.5     | 8.8                   | 16.9              | 7           | 5.8           | 16.0  |
> | InstructSAM-GPT4.1| 33.1     | 14.1             | 23.9               | 34.7      | 47.7  | 13.5       | 21.1      | 13.5      | 44.1     | 19                    | 27.2              | 29.4        | 3.3           | 25.0   |
>
> > Out-of-distrubution: You are testing InstructSAM on two further datasets in the appendix. Why are they called “out-of-distribution”? You are using generic models (SAM2 and CLIP) that are most likely trained on these categories as well.
>
> We apologize for the confusion. You are correct that these datasets are not truly "out-of-distribution" for InstructSAM. The term "out-of-distribution" refers to datasets outside the training distribution of open-vocabulary detection models like OWLv2, CASTDet, and LAE-DINO. These models showed significant performance degradation on these datasets, whereas InstructSAM, equipped with more capable foundation models, performed better.
>
> > Task novelty claim: As you clarify in the related work, InstructCDS does not fundamentally differs from existing instruction-oriented tasks and your claim “new suite of tasks” might be a bit overstated.
>
> - **Instruction-Oriented Object Counting and Segmentation:** While we followed the settings of previous work on Instruction-Oriented Object Detection [1]. We add object counting and object segmentation to form a new suite of tasks, in other words we are the first to implement the comprehensive tasks altogether. This is a significant advancement, particularly for object counting, as current benchmarks like CountBenchVQA [2] are limited to single-class counting with explicit class names in the query. In contrast, InstructCDS requires counting based on implicit instructions.
>
> -  **Novel evaluation framework:** We defined F1-score for multi-class object counting, providing a more informative evaluation metric (line 141).
>
> **References**
>
>
> [1] Pi, Renjie, et al. "INS-DetCLIP: Aligning detection model to follow human-language instruction." ​*The Twelfth International Conference on Learning Representations*​. 2024.
>
> [2] Paiss, Roni, et al. "Teaching clip to count to ten." ​*Proceedings of the IEEE/CVF International Conference on Computer Vision*​. 2023.
>
> We thank you for great comments and hope our response clarified the paper's contribution to this community.

---

> > ### Comment · Reviewer_te26 · 2025-08-06
> >
> > I welcome the additional experiments provided by the authors and believe these significantly strengthen the paper. In particular, the broader evaluation on medium- and low-resolution datasets, as well as land-use classes, adds more evidence of InstructSAM’s applicability across diverse remote-sensing scenarios. I would recommend listing the experiments from the appendix in the main paper. E.g. the xBD results would also show some limitations with room for improvment.
> >
> > I also appreciate that the authors clarified that their setting is an improvement over existing instruction-oriented remote sensing tasks. It would be beneficial to explicitly highlight this broader scope in the paper and acknowledge prior works, which will help readers better contextualize the contributions.
> >
> > Overall, the rebuttal addresses my main concerns regarding dataset diversity, task novelty, and real-world applicability, and I believe the revisions improve the clarity and impact of the work.

---

### Official Review · Reviewer_xfUj · 2025-06-29

**Clarity:** 3
**Significance:** 4
**Originality:** 3
**Rating:** 6
**Confidence:** 4

**Summary:**

The paper investigates zero shot object recognition in the remote sensing domain.

It introduces the InstructSAM model, which first using a VLM (Qwen 2.5 or GPT-4o) to predict the categories and number of objects for each category in the scene. The SAM2 model is used to predict object masks in the scene using a regular grid of point prompts (Generating many potential segmentations which are assumed to cover any objects of interest). The next step is to assign a subset of the masks to counted objects from the VLM. The category text and masks are encoded with CLIP and then compared. A binary integer program solves the assignment problem subject to counting constraints.

This method is tested on EarthInstruct, which is created from two existing datasets: NWPU-CHR-10 and DIOR. And defines 3 distinct task settings: open-vocab, open-ended, and open-subclass.

The technique is also evaluated in terms of inference time, and claims to be nearly constant with respect to the number of boxes.

Ablations cover prompt design and model selection. Experiments also cover error analysis.

**Questions:**

1. Have you tried using the predicted object classes from the LVLM in your SAM2 prompt? I'm wondering what the different in scores would be if you primed SAM based on the VLM output rather than just using a grid of prompt points.

2. What is the maximum size of the inputs to the binary integer program? You state that it is solvable efficiently, but this is wrong in general (unless P=NP). I think it is important to justify that the problem size won't become pathological - or if it does, state the limit at which that will happen. The PuLP solver is good, but it doesn't scale that far.

3. I don't understand why your runtime is constant (wrt to number of boxes), but others are not. Why does inference time increase with the number of objects in instruction-oriented models?

**Ethical Concerns:**

["NO or VERY MINOR ethics concerns only"]

**Final Justification:**

I'm raising my score due to the addition of new experiments especially the one that shows this does NOT work on SAR. I believe this paper is a significant step forward in how we in the vision and AI community think about using using these models to solve problems, and by showing the limitations of where it does not work, it lays groundwork for more progress to be made.

**Limitations:**

yes

**Quality:**

3

**Strengths And Weaknesses:**

Strength:

* The construction of the EarthInstruct dataset with the awareness of not counting objects in an image subject to spatial resolution is well thought out.

* The model selection ablation is informative as to how performance varies as a function of the backend models.

* The mask-label ablation is impressive. If I'm reading this right, then if you just filter boxes by confidence score the version with the BIP matcher outperforms all operating points.

* Combining existing trained models in a novel way to perform well on a zero shot problem is both impressive and practical.


Weakness:

* The prompt design section of the ablation study is unclear. I don't understand what is being varied.

* While the mask-label ablation is very interesting, I think the reason that BIP matching is able to outperform all operating points it that SAM2 is prompted with a regular grid of points, producing many more candidates than needed. The process isn't trained to produce the correct boxes around all objects, its being asked to produce boxes where there may not be a valid object.

* Being confidence free isn't necessarily a good thing. If the final boxes were assigned a score, the ability to filter / prioritize them could be beneficial for some applications. Can this method be imbued with a confidence score?


Comments:

A recent paper: Vision Language Models are Biased (Vo et al 2025) https://arxiv.org/abs/2505.23941
showed a strong bias and reliance on context cues in counting VLMs. This is
recent, so I'm not counting it against the paper, but followup work should
consider this.


[Line 66] Why is bounding box generation with qwen so slow?

[Line 77] Is confidence free actually a pro? Could that also be a con due to no ability to filter based on a score? Can your method be imbued with a confidence score?

[Line 160] Strength: The method for computing AP without confidence values is correct.

[Line 163] Using the maximum F1 might not be fair. It is a "best possible" view and cannot be used to say a candidate model is "better" than another.

[Line 171] Performing a random sample on the selected pairs at cosine similarity > 0.95 and then quantifying the error rate would be valuable to justify this threshold choice.

[Line 201] The 1.2x scale factor seems arbitrary, was it tuned? Not not a big deal if it wasn't.

[Line 222] Clarify, I find this wording hard to follow: For open subclass setting, we set two parent class "means of transport" and "sports field".

[Figure 6] Is the Yellow line adding to the key takeaway of this plot? I'm also confused how mAPnc of InstructSAM w/o matching has different values at different thresholds. Am I missing something here? Please double check these numbers and if the takeaway is that BIP matching outperforms all operating points, state that explicitly or add clarifying text so readers don't make that mistake.

---

> ### Author Rebuttal · Authors · 2025-07-31
>
> **Strengths**
>
> > The mask-label ablation is impressive. If I'm reading this right, then if you just filter boxes by confidence score the version with the BIP matcher outperforms all operating points.
>
> Thank you for recognizing this. Indeed, Figure 6 compares the per-class F1-score of InstructSAM with filtering solely by a single confidence score across all images and categories. Each category requires a different optimal threshold, making a single threshold ineffective across classes. In contrast, the counting-constrained matching process dynamically assigns labels without relying on a fixed threshold, ensuring optimal assignments for each image.
>
> **Weakness**
>
> > The prompt design section of the ablation study is unclear. I don't understand what is being varied
>
> We appreciate your feedback. The "Add Instr" column in Table 4 refers to additional instructions, specifically dataset-specific annotation rules (e.g., NWPU’s “harbor” as distinct piers, excluding small vehicles). See lines 645–651 and 678–693 for details. These instructions guide the LVLM counter to better align with user intent, significantly reducing inherent biases.
>
> > While the mask-label ablation is very interesting, I think the reason that BIP matching is able to outperform all operating points is that SAM2 is prompted with a regular grid of points, producing many more candidates than needed. The process isn't trained to produce the correct boxes around all objects, its being asked to produce boxes where there may not be a valid object.
>
> This is an insightful observation. Indeed, SAM2 often generates more mask proposals than the actual number of objects in the image. Constraint Equation (4) at Line 204 guarantees that all mask proposals are assigned when they are fewer than the predicted object counts.
>
>
> > Being confidence free isn't necessarily a good thing. If the final boxes were assigned a score, the ability to filter / prioritize them could be beneficial for some applications. Can this method be imbued with a confidence score?
>
> Great point again! Each matched mask-label pair in InstructSAM already has a semantic similarity score, $s_{ij}$, which is automatically saved in the final predictions. This score can be used for filtering or prioritization, similar to conventional detectors. For example, in the open-subclass setting, threshold filtering based on $s_{ij}$ improved box mF1 by up to 8.4%.
>
> **Comments and Questions**
>
> > A recent paper: Vision Language Models are Biased (Vo et al. 2025) showed a strong bias and reliance on context cues in counting VLMs. Followup work should consider this.
>
> Thank you for pointing this out. This paper focuses on counterfactual scenarios (e.g., counting legs on a three-legged chicken). In remote sensing, annotation rules may vary across datasets but rarely contradict common sense. Adding annotation rules significantly improves counting accuracy. For example, when distinguishing "bridge" from "overpass," the model often recognizes overpasses as bridges due to their hierarchical relationship. However, annotation rules improved bridge accuracy and overpass recall by large margins:
>
> | Prompt Type| Bridge (Acc / Recall / F1)| Overpass (Acc / Recall / F1) |
> | - | -- | -- |
> | w/o annotation rules | 44.1 / 44.5 / 44.3| 75.9 / 16.7 / 27.4|
> | w/ annotation rules  | 66.3 / 50.3 / 57.2| 63.4 / 64.9 / 64.2|
>
> > [Line 66] Slow Qwen bounding box generation. Why your runtime is constant (wrt to number of boxes), but others are not. Why does inference time increase with the number of objects in instruction-oriented models?
>
> Representing every bounding boxes as text requires more tokens. Experimental results show that inference time is proportional to the number of output tokens. In contrast, counting requires far fewer tokens, making InstructSAM-Qwen faster than Qwen2.5-VL when the number of predicted objects exceeds five.
>
> > [Line 77] Is confidence free actually a pro? Could that also be a con due to no ability to filter based on a score?
>
> Confidence-free methods are advantageous in zero-shot or OOD scenarios, where bounding box quality is highly sensitive to threshold selection. However, if a validation dataset is available for tuning thresholds, confidence scores can be useful.
>
> > [Line 163] Using the maximum F1 might not be fair. It is a "best possible" view and cannot be used to say a candidate model is "better" than another.
>
> We agree. A fairer approach is to tune thresholds on training data. InstructSAM can also be evaluated for "best possible" performance by enabling confidence filtering, as shown in the table above.
>
> The impact of threshold filtering varies across different settings. For tasks like OVD and OED, which involve a large number of categories, a single threshold is ineffective at balancing performance across all classes. However, in scenarios with fewer classes, such as subclass-specific tasks, threshold filtering becomes more effective and can significantly improve performance.
>
> | Model| Threshold filtering | OVD    | OED | OSD/sports field | OSD/transport |
> | --- | --- | --- | --- | --- | --- |
> | Qwen2.5-VL        | ×| 36.4| 32.0| 32.4| 42.2|
> | InstructSAM-Qwen  | ×| 38.9| 29.6| 33.5| 41.9|
> | InstructSAM-GPT4o | ×| 41.8| 31.3| 46.9| 44.2|
> | GPT-4o+OWL        | √| 37.4| 24.0| 19.8| 65.9|
> | InstructSAM-Qwen  | √| 38.9| 29.8 (0.2↑) | 34.2 (0.7↑)| 48.9 (7↑)|
> | InstructSAM-GPT4o | √| 41.9 (0.01↑) | 31.3| 49.3 (2.4↑)| 52.6 (8.4↑)  |
>
> > [Line 160] Strength: The method for computing AP without confidence values is correct.
>
> Thanks, we hope this method can raise the awareness of fair comparison bewteen confidence-based and confidence-free methods in generic object detection domain.
>
> > [[Line 171] Performing a random sample on the selected pairs at cosine similarity > 0.95 and then quantifying the error rate would be valuable to justify this threshold choice.
>
> Thank you for the suggestion. We manually evaluated open-ended counting results on NWPU using Qwen2.5-VL and GPT-4o. The error rate was 3% (1 out of 32), with the only error being "sports\_field" mapped to "baseball\_field." Most synonyms were correctly mapped, but some misses (e.g., "tank" vs. "storage tank") highlight the need for better RS-specific CLIP models.
>
> > [Line 201] The 1.2x scale factor seems arbitrary, was it tuned? Not not a big deal if it wasn't.
>
> It was not tuned initially. We hypothesized that adding context would improve accuracy since CLIP is trained on full images rather than object crops. Experiments on NWPU in the OVD setting confirmed this, with 1.3x yielding the best results:
>
> | Model| Crop Scale | Box AP |
> | -- | --- | -- |
> | InstructSAM-GPT4o | 1.1×| 41.0|
> | InstructSAM-GPT4o | 1.2x| 41.8|
> | InstructSAM-GPT4o | 1.3x| 42.7|
> | InstructSAM-GPT4o | 1.4x| 41.9|
>
> > [Line 222] Clarify, I find this wording hard to follow: For open subclass setting, we set two parent class "means of transport" and "sports field".
>
> The subclasses for "means of transport" and "sports field" are listed in Table 7 (Appendix B). We will add a reference to this table in the main text for clarity.
>
> > [Figure 6] Is the Yellow line adding to the key takeaway of this plot? I'm also confused how mAPnc of InstructSAM w/o matching has different values at different thresholds. Am I missing something here? Please double check these numbers and if the takeaway is that BIP matching outperforms all operating points, state that explicitly or add clarifying text so readers don't make that mistake.
>
> The yellow line represents InstructSAM without matching, where predictions are filtered by a similarity score threshold before evaluation. That's the reason the value differs. We will clarify this in the revised manuscript. The key takeaway is that BIP matching outperforms all operating points, and we will explicitly state this to avoid confusion.
>
> > Have you tried using the predicted object classes from the LVLM in your SAM2 prompt? I'm wondering what the difference in scores would be if you primed SAM based on the VLM output rather than just using a grid of prompt points.
>
> The official implementations of SAM and SAM2 do not support text prompts, though this capability was discussed in the original paper [1]. Additionally, prior work on text-prompted segmentation for remote sensing (e.g., GeoPixel [2]) has shown limited generalization to classes outside the training set (see Figure 10, Appendix G.2). For these reasons, we use gridded point prompts in our experiments.
>
> > What is the maximum size of the inputs to the binary integer program? You state that it is solvable efficiently, but this is wrong in general (unless P=NP). I think it is important to justify that the problem size won't become pathological - or if it does, state the limit at which that will happen. The PuLP solver is good, but it doesn't scale that far.
>
> Thank you for this insightful question. You are correct that BIP is NP-hard in general; our claim refers to it being **tractable in practice** for the problem scales encountered in our framework.
>
> In our experiments, the problem size was modest. For example, on NWPU, the maximum number of mask proposals (N) was 200 and categories (M) was 8, with the solver finding an optimal solution in under 0.07s.
>
> To further justify this, a multiple linear regression of the solution time against N and M yielded the equation: `Time = -0.0052 + 0.00013*N + 0.00535*M`. This model's high coefficient of determination (R² = 0.84) confirms that the solution time scales predictably and near-linearly within our operational range, ensuring practical efficiency.
>
> **References**
>
> [1] Kirillov, Alexander, et al. "Segment anything." CVPR. 2023.
>
> [2] Shabbir, Akashah, et al. "Geopixel: Pixel grounding large multimodal model in remote sensing." ICML. 2025.
>
> We thank you for great comments and hope our response clarified the paper's contribution to this community.

---

> > ### Comment · Reviewer_xfUj · 2025-08-02
> >
> > Thank you for the clarifications and providing interesting new experiments. Noting how the CLIP scores can serve as additional scoring and filtering (for cases where high precision is required) addresses one of my main concerns.
> >
> > Similarly I'm satisfied with the explanation of why BIP can be considered efficient in this case, but I strongly recommend the authors revise the paper to make that clear in the main text.
> >
> > Regarding points raised by other reviewers about SAR and multispectral imagery: while this paper focuses on the preprocessed RGB domain, which I believe is sufficient for a solid accept, expanding to other modalities is an obvious and exciting direction. I am unsurprised that the current formulation struggles with SAR imagery, given SAM2's likely lack of SAR training data. However, I believe this work lays important groundwork for future adaptations-such as replacing the SAM2 backbone with a foundation model trained jointly on ground and RS imagery, or incorporating modality adaptation layers as in [1]-even if such approaches require training.
> >
> > With the addition of the new experiments (especially the case demonstrating that this approach does NOT work on SAR), I am considering raising my rating to a strong accept.
> >
> > The training-free formulation here is both scientifically novel and methodologically sound, and I would be willing to defend this paper if its acceptance is in question.
> >
> > One follow up question:
> >
> > Does the paper mention which PuLP backend was used? Last time I checked (many years ago) there was a substantial difference between the free and non-free solvers. The accessibility would be limited if only the non-free solvers were efficient enough. I would also recommend searching for the N and M at which the problem becomes intractable, which will help set expectations and guide future applications of this work.
> >
> > [1] Xiong, Zhitong, et al. "Neural plasticity-inspired multimodal foundation model for earth observation." arXiv preprint arXiv:2403.15356 (2024).

---

> > > ### Author Response · Authors · 2025-08-03
> > >
> > > We sincerely appreciate your positive feedback on our training-free framework and the additional experiments. Regarding your follow-up questions:
> > >
> > > 1. Solver Details: We used the open-source CBC (Coin-or Branch and Cut) solver. For all problem sizes in our study, solutions were obtained efficiently (<0.07s) via the Feasibility Pump algorithm.
> > >
> > > 2. Scalability Analysis: We extended our tests to larger problem sizes (N=1500 masks, M=40 categories). The scaling relationship remained consistent with our original findings:
> > > `Time = -1.184 + 0.0014*N + 0.0593*M` (R²=0.88).
> > > The maximum solving time at this scale was 3.9s, demonstrating practical efficiency for foreseeable applications.
> > >
> > > We will incorporate these implementation details in the paper revision, as you suggested. Thank you for this constructive feedback.

---

### Official Review · Reviewer_apuM · 2025-06-30

**Clarity:** 3
**Significance:** 3
**Originality:** 3
**Rating:** 5
**Confidence:** 3

**Summary:**

The paper introduces InstructSAM, a training-free framework for instruction-driven object recognition in remote sensing imagery. The authors first propose InstructCDS, a suite of tasks covering open-vocabulary, open-ended, and open-subclass scenarios for object counting, detection, and segmentation. To benchmark these tasks, they construct EarthInstruct and incorporate dataset-specific annotation rules (e.g., excluding low-resolution vehicles). InstructSAM leverages three components: An LVLM to interpret instructions and predict object categories/counts. SAM2 to generate class-agnostic mask proposals. CLIP (domain-tuned) to compute semantic similarity, followed by binary integer programming (BIP) to assign categories to masks under counting constraints. Experiments show InstructSAM matches or surpasses specialized baselines, maintains near-constant inference time regardless of object count, and reduces output tokens by 89%. The framework eliminates task-specific training and confidence thresholds, enhancing scalability and robustness.

**Questions:**

1. Scalability with Larger LVLMs: InstructSAM uses GPT-4o (proprietary) and Qwen2.5-VL (open). How does it affect the performance with other large open LVLMs (e.g., LLaMA-3-70B, Gemini 1.5)?

2. SAM2 Alternatives: SAM2 is central to mask proposals. Did you explore other lightweight alternatives (e.g., MobileSAM, FastSAM) for edge deployment? Also quantify trade-offs (speed vs. recall)?

**Ethical Concerns:**

["NO or VERY MINOR ethics concerns only"]

**Final Justification:**

While some of my key concerns have been addressed, I improve my original borderline accept rating.

**Limitations:**

Yes

**Quality:**

3

**Strengths And Weaknesses:**

Strengths:
1. Strong empirical results across diverse tasks. InstructSAM-GPT4o outperforms remote sensing-specific models. 2. Ablations validate design choices (e.g., prompt engineering, model scaling). 3. Error analysis reveals clear failure modes (e.g., background confusion due to CLIP’s scene-level bias).

Weakness:
1) Reliance on pre-trained models (SAM2, CLIP, LVLMs) introduces inherited biases. Sometimes SAM2 struggles with complex geometries and only provide patches in these complex geometries.
2) Limited validation on some goal-oriented instruction prompts. For example, given the prompts "detect flooded buildings", it's hard to find flooded building validation dataset. Therefore its hard to validate model performance for such kind of prompt.

---

> ### Author Rebuttal · Authors · 2025-07-31
>
> **Weekness**
>
> > Reliance on pre-trained models (SAM2, CLIP, LVLMs) introduces inherited biases. Sometimes SAM2 struggles with complex geometries and only provide patches in these complex geometries.
>
> We acknowledge this limitation, as it is an inherent drawback of using a generic SAM2 for segmenting EO images. Recent research has developed SAM variants tailored for EO imagery [1, 2], though their pre-trained weights are not yet publicly available. We believe InstructSAM will benefit from any pre-trained models (SAM2, CLIP, LVLMs) models for EO images, thanks to its training-free design. This adaptability ensures that future advancements in EO-specific foundation models can be seamlessly integrated into our framework.
>
> > Limited validation on some goal-oriented instruction prompts. For example, given the prompts "detect flooded buildings", it's hard to find flooded building validation dataset. Therefore, it's hard to validate model performance for such kind of prompt.
> > We agree the evalution of goal-oriented instruction prompts would be valuable. Here we further tested InstructSAM on FloodNet [3] (0.02m GSD, incorporating 9 classes covering diverse object of interests in remote sensing community). The LVLM is the key to understanding goal-oriented instructions. With structured prompt design and given the annotation rules, InstructSAM counts objects at a very high accuracy, while SegEarth-OV and OWLv2 failed to comprehend the subtle difference between flooded buildings and non-flooded buildings. However, given the clear texture presented in such high-definition drone images, SAM2 may segment an object into patches, reducing the recall of mask proposals and detection performance. This is consistent with the limitations we found in Appendix E.
>
> | model              | Metric | Building-flooded | Building-non-flooded | Pool | Vehicle | Road-flooded | Road-non-flooded | Water | Tree | Grass | mIoU/F1 |
> | -------------------- | -------- | ------------------ | ---------------------- | ------ | --------- | -------------- | ------------------ | ------- | ------ | ------- | --------- |
> | SegEarth-OV        | IoU    | 0                | 37.5                 | 19   | 13      | 8.8          | 51.6             | 53.7  | 55   | 77.6  | 35.1    |
> | InstructSAM-GPT4.1 | IoU    | **13.8**             | 36.7                 | 14.9 | 15.7    | 9.2          | 19.6             | 44.3  | 36   | 61.6  | 28      |
> | OWLv2              | Cnt F1 | 11.8             | 48.7                 | 64.5 | 78.5    | -            | -                | -     | -    | -     | 50.9    |
> | InstructSAM-GPT4.1 | Cnt F1 | **78.5**             | **71.3**               | **84.2** | **80.6**    | -            | -                | -     | -    | -     | 78.7    |
> | OWLv2              | Det F1 | 0.8              | 24                   | 37.1 | 29.2    | -            | -                | -     | -    | -     | 22.8    |
> | InstructSAM-GPT4.1 | Det F1 | **2.8**              | 5.8                  | 22.2 | 6.5     | -            | -                | -     | -    | -     | 9.32    |
>
> **Questions**
>
> > Scalability with Larger LVLMs: InstructSAM uses GPT-4o (proprietary) and Qwen2.5-VL (open). How does it affect the performance with other large open LVLMs (e.g., LLaMA-3-70B, Gemini 1.5)?
>
> We evaluated InstructSAM with other large open LVLMs, such as Gemma 3 and Mistral-Small, to assess scalability of InstructSAM. These results show that other open LVLMs, such as Gemma 3 and Mistral-Small, also achieve competitive performance. Notably, Qwen2.5-VL-72B significantly outperformed its 7B counterpart, demonstrating good scalability with larger models.
>
> | Model                 | Counting mAcc | Counting mRecall | Counting mF1 |
> | ----------------------- | --------------- | ------------------ | -------------- |
> | Gemma3-12B            | 73.2          | 76.4             | 72.8         |
> | Mistral-Samll-3.2-24B | 84.5          | 65.6             | 70.8         |
> | Qwen2.5-VL-7B         | 81.9          | 70.5             | 73.2         |
> | Qwen2.5-VL-72B        | 81.2          | 83.9             | 79.5         |
>
> > SAM2 Alternatives: SAM2 is central to mask proposals. Did you explore other lightweight alternatives (e.g., MobileSAM, FastSAM) for edge deployment? Also quantify trade-offs (speed vs. recall)?
>
> Thank you for this valuable suggestion. We evaluated several variants of SAM2 to analyze the trade-offs between model size, inference time per image, and mask proposal recall. SAM2 Small strikes a good balance between model size and recall, making it a viable option for resource-constrained scenarios.
>
> | Model      | Size (M) | Inference Time (s) | Mask Proposal Recall |
> | ------------ | ---------- | -------------------- | ---------------------- |
> | SAM2 Large | 224.4    | 3.52               | 82.4                 |
> | SAM2 Base+ | 80.8     | 3.38               | 81.4                 |
> | SAM2 Small | 46       | 3.11               | 79.1                 |
> | SAM2 Tiny  | 38.9     | 3.01               | 74.1                 |
>
> **References**
>
> [1] Yan, Zhiyuan, et al. "RingMo-SAM: A foundation model for segment anything in multimodal remote-sensing images." *IEEE Transactions on Geoscience and Remote Sensing*. 2023.
>
> [2] Shan, Zhe, et al. "ROS-SAM: High-quality interactive segmentation for remote sensing moving object." *Proceedings of the Computer Vision and Pattern Recognition Conference*. 2025.
>
> [3] Maryam, Rahnemoonfar et al. "FloodNet: A High Resolution Aerial Imagery Dataset for Post Flood Scene Understanding" *IEEE Access*. 2021.
>
> We thank you for great comments and hope our response clarified the paper's contribution to this community.

---

### Official Review · Reviewer_QWiC · 2025-07-02

**Clarity:** 3
**Significance:** 3
**Originality:** 3
**Rating:** 5
**Confidence:** 3

**Summary:**

The authors formulate a new set of problems for Remote Sensing i.e Instruction-Oriented Object Counting, Detection, and Segmentation in open-vocabulary, open-ended, and open-subclass settings. They then proceed by providing the first benchmark on this suite of problems named Earth-Instruct built on top of two existing datasets. Finally, they propose a framework that excels in these tasks build upon existing large models like GPT-4o and SAM2.

**Questions:**

I believe the main drawback of this work is the diversity of the benchmark itself both in terms of the classes it encompasses as well as in terms of its input --> Sensors, domains of interest (land, marine etc.) , Ground sampling distances etc.

- How were these datasets selected? The explanation given in L129 is a bit vague.
- How can the benchmark be (potentially) extended?
- Extending the evaluation of InstructSAM to standard RS segmentation datasets outside of the benchmark and comparing its performance with standard baselines would be interesting and could help us identify its current limits.
  - For example, how would it perform on pixel based land cover land use classification e.g on FLAIR[1].
  - What is the impact of the granularity of the target classes?
  - How does it adapt to varying environmental conditions?
- Some experiments on other popular EO modalities like SAR and Multispectral data could showcase the current limitations of adapting this training-free methodology. (Example SAR segmentation dataset for flood mapping: Kuro Siwo [2])

[1] Garioud, Anatol, et al. "FLAIR: a country-scale land cover semantic segmentation dataset from multi-source optical imagery." Advances in Neural Information Processing Systems 36 (2023): 16456-16482.

[2] Bountos, Nikolaos Ioannis, et al. "Kuro Siwo: 33 billion $ m^ 2$ under the water. A global multi-temporal satellite dataset for rapid flood mapping." Advances in Neural Information Processing Systems 37 (2025): 38105-38121.

**Ethical Concerns:**

["NO or VERY MINOR ethics concerns only"]

**Final Justification:**

The authors addressed all of my concerns and even introduced new experiments, further solidifying their paper. The authors promised to extend the limitations discussion including the sensitivity to target class granularity, environmental variability, and performance constraints of new modalities e.g SAR, improving the reader's understanding of the scope of this work. Overall this is a good paper, and I stick to my initial recommendation for acceptance.

**Limitations:**

The authors address the limitations of their work in the appendix.

**Paper Formatting Concerns:**

I do not see any violations of the paper formatting.

**Quality:**

3

**Strengths And Weaknesses:**

**Strenghts**
The authors introduce a novel set of problems, along with a respective benchmark to complement the tasks. Their benchmark is built on top of 2 diverse detection datasets. The authors propose a novel, training-free, method that performs well on these tasks setting a strong baseline for future methods. The out-of-the-box performance of InstructSAM is particularly interesting. Overall the paper is clear and well written.


**Weaknesses**

***Minor Weaknesses***

Class legends in figures are very small. One has to zoom in a lot to actually read them

***Major Concerns***



- The benchmark is built upon two pre-existing RGB and RGB+NIR datasets, at varying Ground Sampling Distances (GSDs) spanning a few cm to 30m. The included classes are very specific and do not cover major areas of interest in the Earth Observation community e.g land use land cover, crops, floods, burnt areas, tree species. Many of these categories have hierarchical structures e.g forest type, tree genus, tree species etc often resulting to finegrained classification tasks. Solving them would typically require varying sensors and Ground Sampling Distances as input. Including more diverse data could make the benchmark more representative for EO. Nevertheless this work is an important first step towards this direction.
- It is not very clear why the benchmark was limited to these two datasets. Obviously, there are not many detection datasets for Earth Observation but there are definitely more than two. Even more, segmentation datasets could be transformed to detection tasks, incorporating greater class and GSD diversity
- Relying on SAM2 for mask proposal may be restricting when switching to sensors highly different from optical images (e.g Synthetic Aperture Radar (SAR) and Interferometric SAR (InSAR)). In such cases the benefits of the proposed methodology could diminish completely. An experiment studying this could add value to the paper.

---

> ### Author Rebuttal · Authors · 2025-07-31
>
> **Minor Concerns:**
>
> > Class legends in figures are very small. One has to zoom in a lot to actually read them.
>
> Thank you for pointing this out. We will improve the display of legends in the final version.
>
> **Major Concerns and questions:**
>
> > The included classes are very specific and do not cover major areas of interest in the Earth Observation community e.g land use land cover, crops, floods, burnt areas, tree species. Many of these categories have hierarchical structures e.g forest type, tree genus, tree species etc often resulting to finegrained classification tasks. Solving them would typically require varying sensors and Ground Sampling Distances as input. Including more diverse data could make the benchmark more representative for EO. Nevertheless, this work is an important first step towards this direction.
>
> Thank you for acknowledging our contribution as an important first step. Our primary focus is on object recognition tasks, but we agree that extending InstructSAM to segmentation datasets is valuable for addressing broader EO challenges. To this end, we have conducted additional experiments on FloodNet and FLAIR datasets to explore its potential for land cover and flood segmentation tasks.
>
> > How were these datasets selected? The explanation given in L129 is a bit vague.
>
> We appreciate the opportunity to clarify our dataset selection process:
>
> * **Diversity in resolution and modality:** Current object detection datasets in remote sensing are predominantly high-resolution and RGB. NWPU-VHR-10 includes NIR bands, while DIOR spans a wide range of resolutions (0.3m to 30m), representing diverse GSDs.
> * **Annotation rules and user requirements:** NWPU-VHR-10 is a subset of DIOR in terms of classes, but their annotation rules differ significantly, reflecting varying user requirements. For example, NWPU excludes vehicles in low-resolution images, while DIOR includes them. This makes these datasets ideal for evaluating a model's ability to follow user-specific instructions beyond simple category matching.
> * **Comparability with prior work:** Both datasets have been widely used in previous zero-shot object detection and segmentation studies, such as ZoRI (AAAI'25) and DesReg (AAAI'24). We provide a detailed comparison with these methods in Appendix D1, Table 10.
>
> > It is not very clear why the benchmark was limited to these two datasets.  Obviously, there are not many detection datasets for Earth Observation but there are definitely more than two.
>
> We agree that there are other datasets available. However, we have found that NWPU-VHR-10 and DIOR are sufficient to demonstrate both the strengths and limitations of InstructSAM in optical object recognition:
>
> * **Strengths:** InstructSAM excels at following diverse instructions and operates in a training-free paradigm.
> * **Limitations:** Its performance is constrained by the capabilities of SAM and CLIP models, as detailed in Appendix E.
>
> That said, we acknowledge the value of extending the benchmark to other modalities and tasks. This is an exciting direction for future work.
>
> > How can the benchmark be (potentially) extended?
>
> The benchmark can be extended in several ways:
>
> * **Increase in class diversity:** Adding more fine-grained categories, such as tree species or specific aircraft types.
> * **Incorporate additional modalities:** Including SAR data to evaluate cross-modality generalization.
> * **Expand task scope:** Extending to land cover classification and change detection tasks.
>
> > Extending the evaluation of InstructSAM to standard RS segmentation datasets outside of the benchmark and comparing its performance with standard baselines would be interesting and could help us identify its current limits. For example, how would it perform on pixel based land cover land use classification e.g on FLAIR.
>
> Thank you for this suggestion. To adapt InstructSAM for land cover tasks, we made the following key modifications:
>
> 1. **Area-based prediction:** Since segmentation tasks involve "uncountable" objects, we used the LVLM to predict the proportion of each class in the image (e.g., 30% building, 70% brushwood).
> 2. **Modified objective function:** We added a penalty term to minimize the difference between the predicted and assigned areas for each class. The updated objective function is:
>
> $$
> \min_{X} \frac{1}{N} \sum_{i=1}^N \sum_{j=1}^M (1 - s_{ij}) x_{ij} + \lambda \sum_{j=1}^M \left| \sum_{i=1}^N a_i x_{ij} - t_j \right|
> $$
>
> $$
> s.t. \sum_{j=1}^M x_{ij} = 1, \quad \forall i \in {1, ..., N}
> $$
>
> where $t_j$ is the target area for class $j$ predicted by LVLM, $a_i$ is the area of mask $i$, and $\lambda$ is the weight of the penalty term.
>
> The zero-shot performance on **FLAIR** dataset was promising. InstructSAM-GPT4.1 outperformed the SegEarthOV baseline. The penalty significantly improved the accuracy of mask-label assignments.
>
> | Model                 | building | pervious surface | impervious surface | bare soil | water | coniferous | deciduous | brushwood | vineyard | herbaceous vegetation | agricultural land | plowed land | swimming pool | mIoU |
> | ----------------------- | ---------- | ------------------ | -------------------- | ----------- | ------- | ------------ | ----------- | ----------- | ---------- | ----------------------- | ------------------- | ------------- | --------------- | ------ |
> | SegEarth-OV [1]       | 47.6     | 6.5              | 0.1                | 13.7      | 49.6  | 10.6       | 3.1       | 4.2       | 34.5     | 8.8                   | 16.9              | 7           | 5.8           | 16   |
> | InstructSAM ($λ=0$)    | 25.8     | 9                | 4.4                | 28.8      | 45.3  | 11.4       | 21.7      | 14.1      | 46.9     | 10.9                  | 22.7              | 28.4        | 2.5           | 20.9 |
> | InstructSAM ($λ=0.05$) | 33.1     | 14.1             | 23.9               | 34.7      | 47.7  | 13.5       | 21.1      | 13.5      | 44.1     | 19                    | 27.2              | 29.4        | 3.3           | 25   |
> | InstructSAM ($λ=0.1$)  | 32.2     | 13.7             | 23.5               | 34.7      | 46.4  | 13.6       | 20.9      | 13.6      | 43.9     | 18.9                  | 26.8              | 29          | 3.3           | 24.6 |
>
> > The impact of the granularity of the target classes?
>
> InstructSAM performs well for coarse-grained categories (e.g., "means of transport") but struggles with fine-grained distinctions (e.g., Boeing 777 vs. Airbus A320, or hockey field vs. soccer field). This limitation arises because current RS-specific CLIP models (e.g., RemoteCLIP, GeoRSCLIP, SkyCLIP) are not trained on semantically rich datasets. Incorporating high-quality, diverse training data (e.g., from OpenStreetMap tags) could significantly enhance their capabilities.
>
> > How does it adapt to varying environmental conditions?
>
> This is an excellent point. The appearance of objects can vary significantly across geolocations and seasons. Adding geolocation and seasonal information to the prompt could improve instruction-oriented object counting. We plan to explore this in future work.
>
> > Extending the evaluation of InstructSAM to standard RS segmentation datasets outside of the benchmark and comparing its performance with standard baselines would be interesting and could help us identify its current limits.
>
> We conducted additional experiments on the SARDet-100k [2] dataset to evaluate InstructSAM's performance on SAR imagery. These results highlight the challenges of applying current LVLMs and SAM2 to SAR data. Generic LVLMs struggle to interpret SAR imagery, and SAM2 fails to generate meaningful masks. This underscores the need for modality-specific adaptations in future work.
>
> | Model              | Counting mF1 | Object Recall |
> | -------------------- | -------------- | --------------- |
> | InstructSAM-Qwen   | 0.3          | 1.1           |
> | InstructSAM-GPT4.1 | 6.0          | 1.1           |
>
> **References:**
>
> [1] Li, Kaiyu, et al. "SegEarth-OV: Towards training-free open-vocabulary segmentation for remote sensing images." ​*Proceedings of the Computer Vision and Pattern Recognition Conference*​. 2025.
>
> [2] Li, Yuxuan, et al. "SARDet-100k: Towards open-source benchmark and toolkit for large-scale sar object detection." *Advances in Neural Information Processing Systems.* 2024.
>
> We thank you for great comments and hope our response clarified the paper's contribution to this community.

---

> > ### Comment · Reviewer_QWiC · 2025-08-05
> > **Response to the authors rebuttal**
> >
> > Overall, I am pleased with the authors’ rebuttal. They have addressed all of my concerns and have supported their responses with new experiments where appropriate. The performance degradation in SAR imagery is expected. The authors should consider discussing it in the manuscript.
> >
> > The limitations of this work are clear. Potentially augmenting the limitations discussion with the impact of target granularity and the robustness to shifts in environmental conditions would improve the reader's understanding of the scope of this work. However, there is a straightforward and multi-dimensional roadmap for future work, which strengthens the paper's overall contribution including benchmark extension, geolocation information, utilization of RS-specific foundation models.
> > I will stick to my initial assessment recommending acceptance.

---

> > > ### Author Response · Authors · 2025-08-06
> > >
> > > We thank the reviewer for their constructive assessment and recognition of our revisions.
> > >
> > > Per the recommendation, we will augment the Limitations section to explicitly address:
> > > (i) performance sensitivity to target class granularity,
> > > (ii) robustness challenges under environmental variability (geolocation/seasonality), and
> > > (iii) SAR modality performance constraints.
> > >
> > > Concurrently, we will include the future work directions on RS-specific foundation models to address these issues.
> > >
> > > These additions will provide clearer contextual boundaries for our contributions and highlight a multi-dimentional roadmap for future work. We will implement these edits in the camera-ready version.

---

### Note · Authors · 2025-08-12

We sincerely thank the reviewers for their constructive feedback and recognition of our contributions. We appreciate their acknowledgement that the proposed InstructCDS task and InstructSAM framework represent an important first step for instruction-oriented remote sensing object recognition, and that our rebuttal has addressed the major concerns with additional experiments and clarifications.

To better demonstrate the scope and adaptability of InstructSAM, we extended the evaluation beyond the original NWPU and DIOR datasets:

* ​**Increased GSD range**​: from **0.02 m** (FloodNet) to **30 m** (DIOR), ensuring robustness across resolutions.
* ​**Expanded sensor modalities**​: **RGB**, **NIR** (NWPU), and **SAR** (SARDet), enabling cross-modality assessment.
* ​**Diverse acquisition platforms**​: **UAV** (FloodNet), **crewed aircraft** (NWPU, FLAIR), and **satellites** (NWPU, DIOR, SARDet).
* ​**Broader EO area of interests**​: **floods** (FloodNet), **building detection** (xBD), **land cover** (FLAIR), and **general object detection** (NWPU, DIOR, SARDet).
* ​**Standard land cover datasets**​: InstructSAM showed promising zero-shot performance on FLAIR, indicating its **transferability to standard land cover tasks**.

We also expanded our evaluation methodology to highlight both performance limits and fairness:

* ​**Maximum capability analysis**​: We demonstrated how InstructSAM can benefit from **confidence score filtering** and tested its performance ceiling under optimal conditions.
* ​**Scalability studies**​: We examined open-source LVLMs of varying sizes (Qwen2.5-VL-7B, Qwen2.5-VL-72B, Gemma3-12B, Mistral-Small-24B), as well as lightweight mask proposers, quantifying the trade-offs between accuracy, runtime, and resource footprint.

We have also clarified the novelty of the InstructCDS tasks and EarthInstruct benchmark, emphasizing their comprehensive coverage of instruction settings and recognition tasks. The training-free design of InstructSAM, combined with its efficiency and scalability, establishes it as a significant advancement for instruction-oriented object recognition in earth observation.

We believe the revised manuscript will present a more rigorous evaluation and clearer articulation of our contributions. We are confident that this work will **inspire future research in remote sensing foundation models  to tackle the identified challenges**. Thank you for your time and constructive engagement.

---

### Decision · Program_Chairs · 2025-09-17

**Decision:**

Accept (poster)

**Comment:**

Following the discussion phase, all reviewers recommended acceptance (1 Strong Accept, 3 Accepts), citing the paper's clear writing, novelty, and impressive results. They highlighted the introduction of a new benchmark with potential value to the community, as well as the thorough evaluation. The rebuttal effectively addressed many of their concerns—for example, through the addition of extensive experiments on new datasets, as well as generalization and scaling analyses. Accordingly, the ACs have decided to accept the paper. Please incorporate the reviewers' feedback when preparing the camera-ready version.